# Glial *betaPix* is essential for blood vessel development in the zebrafish brain

**Shihching Chiu[1], Qinchao Zhou[1], Chenglu Xiao[2], Linlu Bai[1], Xiaojun Zhu[1]\*, Wanqiu Ding[1]\*, Jing-Wei Xiong[1,2]\***

[1]Beijing Key Laboratory of Cardiometabolic Molecular Medicine, Institute of Molecular Medicine, College of Future Technology, Peking University, Beijing, China; [2]School of Basic Medical Sciences, The Second Affiliated Hospital, Institute of Biomedical Innovation, The MOE Basic Research and Innovation Center for the Targeted Therapeutics of Solid Tumors, Jiangxi Medical College, Nanchang University, Nanchang, China

## eLife Assessment

This **valuable** article presents findings supported by **solid** data to identify a surprising glia-exclusive function for betapix in vascular integrity and angiogenesis. The article also describes the optimization of a modified CRISPR-based Zwitch approach to generate conditional knockouts in zebrafish.

**\*For correspondence:**
zhuxiaojun@pku.edu.cn (XZ);
dingwq@pku.edu.cn (WD);
jingwei_xiong@pku.edu.cn (J-WX)

**Competing interest:** The authors declare that no competing interests exist.

**Abstract** The formation of blood–brain barrier and vascular integrity depends on the coordinated development of different cell types in the brain. Previous studies have shown that zebrafish *bubblehead* (*bbh*) mutant, which has mutation in the *betaPix* locus, develops spontaneous intracerebral hemorrhage during early development. However, it remains unclear in which brain cells *betaPix* may function. Here, we established a highly efficient conditional knockout method in zebrafish by using homology-directed repair-mediated knockin and knockout technology and generated *betaPix* conditional trap (*betaPix*^ct) allele in zebrafish. We found that *betaPix* in glia, but neither neurons, endothelial cells, nor pericytes was critical for glial and vascular development and integrity, thus contributing to the formation of blood–brain barrier. Single-cell transcriptome profiling revealed that microtubule aggregation signaling stathmins and pro-angiogenic transcription factors *Zfhx3/4* were downregulated in glial and neuronal progenitors, and further genetic analysis suggested that betaPix may act upstream on the PAK1-Stathmin and Zfhx3/4-Vegfaa signaling to regulate glia migration and angiogenesis. Therefore, this work reveals that glial *betaPix* plays an important role in brain vascular development in zebrafish embryos and possibly human cells.

## Introduction

Intracerebral hemorrhage (ICH) is a life-threatening stroke type with the worst prognosis and few proven clinical treatments. Most patients who survive ICH end up with disabilities and are at risk for recurrence, cognitive decline, and systemic vascular issues, making this condition particularly significant among neurological disorders (*Sheth, 2022*; *Chiang et al., 2025*). Several rodent models have been used for modeling ICH, including autologous blood injection, collagenase injection, thrombin injection, and micro-balloon inflation techniques (*Thangameeran et al., 2023*). Zebrafish (*Danio rerio*) exhibit a closed circulatory system and regulatory pathways that are highly conserved among vertebrates (*Isogai et al., 2001*) with relatively cheap and easy-to-maintain disease models. Over the past decades, varieties of zebrafish mutants that spontaneously developed brain hemorrhage have been generated in mutagenesis screens, such as *redhead* (*Buchner et al., 2007*), *reddish* (*Jin et al., 2007*;

*Hegarty et al., 2013*) and *bubblehead* (*bbh*) (*Stainier et al., 1996*; *Chen et al., 2001*). Elucidating mutated gene function in these zebrafish models enables us to gain insights into the genetic basis and pathophysiological mechanisms of ICH development. The *bbh* mutant was identified independently in two large-scale ethylnitrosourea-induced mutagenesis screens (*Stainier et al., 1996*; *Chen et al., 2001*) and positional cloning revealed that p21-activated kinase (Pak)-interacting exchange factor beta (*betaPix*) gene was mutated in *bbh* mutants (*Liu et al., 2007*). *betaPix* contains SH3 domain that binds group I PAKs, and Dbl homology (DH) and pleckstrin homology (PH) domains that function as a RhoGEF to interact with Rac/Cdc42 small GTPases (*Manser et al., 1998*), thus participating in multiple cellular pathways to regulate cell polarity, adhesion, and migration (*Zhou et al., 2016*).

Various studies have reported that brain vessels are highly supported by neurons and glial cells during development and regeneration (*Morimoto et al., 2024*; *Grinchevskaya et al., 2024*). Glial-specific *betaPix* have been proposed to interact with $\alpha v\beta 8$ integrin and the Band 4.1s in glia, which link to multiple intracellular signaling effectors such as *Wnt7a* and *Wnt7b* (*McCarty, 2009a*), but this hypothesis was not experimentally addressed (*McCarty, 2009b*). To this end, *betaPix* has been shown to interact with $\alpha v\beta 8$ integrins and mediate focal adhesion formation and thus modulate cerebral vascular stability in *bbh* zebrafish mutant (*Liu et al., 2012*). Integrins are the main molecular link between cells and the extracellular matrix (ECM) which serves as scaffolds in the perivascular space. Based on a chemical suppressor screening of *bbh* hemorrhages, we have previously reported that a small molecule called miconazole downregulated the pErk-matrix metalloproteinase 9 (Mmp9) signaling to reduce ECM degradation, thus improving vascular integrity (*Yang et al., 2017*). However, how *betaPix* cell-type specifically contributes to vascular integrity and maturation remains unclear. Here, we generated *betaPix* conditional trap (*betaPix^ct*) alleles by using a CRISPR/homology-directed repair (HDR)-based Zwitch method and found that *betaPix* acted mainly in glia to regulate glial and vascular development and integrity via Stathmin and Zfhx3/4 signaling. Therefore, this work provides the first experimental evidence that *betaPix* acts in glia for vascular integrity development, and its functional conservation between zebrafish and human cells may further guide us to decipher the genetic basis of ICH in the future.

## Results

### Generating *betaPix* conditional trap (*betaPix^ct*) allele by an HDR-mediated knockin and knockout method

To study *betaPix* function in a cell-specific manner, we utilized a donor vector with an invertible gene trap cassette via HDR (Zwitch) (*Sugimoto et al., 2017*; *Ogawa et al., 2021*; *Ogawa and Kikuchi, 2024*) for generating *betaPix* conditional trap (*betaPix^ct*) alleles. Zwitch consists of a splice acceptor conjugated to a red fluorescent protein via a 2A self-cleaving peptide, flanked by two LoxP sites and two Lox 5171 sites in reciprocal orientations. Outside this conditional trap cassette, we included a lens-specific enhanced green fluorescence protein $\alpha$-crystallin:EGFP for screening knockout founders, as well as inserted both left and right arms for targeting homologous sequences on the genome. We modified the Zwitch vector by adding a glycine–serine–glycine (GSG) spacer in front of the 2A peptide to enhance cleavage efficiency, and adding universal guide RNA (UgRNA) (*Wierson et al., 2019*) target sequences outside the left and right homologous arms for linearizing donor vector (*Figure 1A*, *Figure 1—figure supplement 1A*). We chose the CRISPR/Cas9 system to generate double-strand breaks (DSBs) on intron 5 of the *betaPix* locus and chose the most efficient guide RNAs based on their efficiency to induce CRISPR indels. We used either long arms such as ~1000 bp or short arms such as 24 bp that located upstream and downstream of the guide RNA sites. We then co-injected the donor vectors with targeting guide RNA, universal guide RNA, and Cas9 mRNA into one-cell-stage embryos. At 4 days post-fertilization (dpf), we selected $\alpha$-crystallin-EGFP-positive embryos ($F_0$) for raising to adulthood and pre-selected $F_0$ founders by examining inheritable EGFP reporter expression of $F_1$ embryos. To confirm the correct homologous recombination in the *betaPix* locus, we identified potential founders by genomic PCR to examine Zwitch insertion in the *betaPix* locus. Precise knock-in genotypes were further verified using Sanger sequencing (*Figure 1B and C*, *Figure 1—figure supplement 1B*). Among 184 adult $F_0$ founders with $\alpha$-crystallin-EGFP expression, we found 84 founders had EGFP-positive $F_1$ embryos, and 7 founders had expected PCR fragments around both 5' and 3' arms, which two founders had correct insertions in the *betaPix* locus by Sanger sequencing

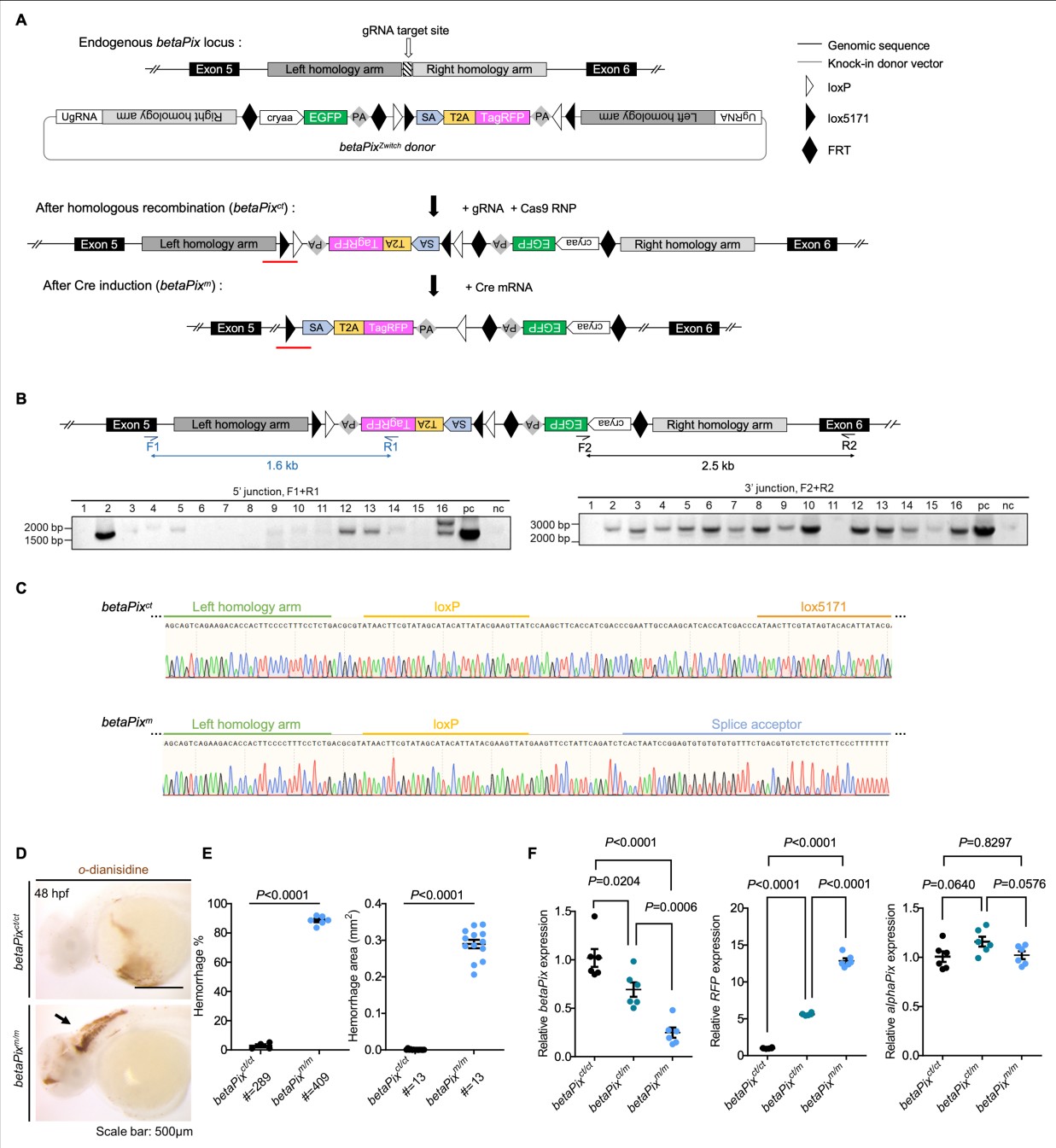

**Figure 1.** Generation of *betaPix* conditional trap (*betaPix^ct^*) allele by a homologous recombination (HDR)-mediated knock-in method. (**A**) Schematic diagram illustrates the HDR-mediated *Zwitch* strategy for generating zebrafish knock-in allele at the *betaPix* locus. (**B**) Genomic PCR analysis of the F₁ embryos confirming the right *Zwitch* insertions. (**C**) Sanger sequencing confirming the junction of *betaPix^ct^* (after HDR-mediated insertion) or *betaPix^m^* (after Cre-mediated inversion) that are highlighted by red lines in (**A**). (**D**) Representative stereomicroscopy images of erythrocytes stained with *o*-dianisidine in *betaPix^ct/ct^* and *betaPix^m/m^* embryos at 48 hpf. Brain hemorrhages, indicated with an arrow, in *Cre* mRNA-injected embryos (*betaPix^m/m^*). Lateral views, anterior to the left. (**E**) Quantification of hemorrhagic parameters in (**D**). Left panel showing hemorrhage percentages, with independent experiments as dots. Right panel showing hemorrhage areas, with each dot representing one embryo, # represents the numbers of embryos scored for each analysis, and three or more individual experiments conducted. Data are presented in mean ± SEM; unpaired Student's *t*-test with individual p-values mentioned in the figure. (**F**) qRT-PCR analysis showing the expression of *betaPix*, *RFP*, and *alphaPix* in *betaPix^ct/ct^*, *betaPix^ct/m^*, and *betaPix^m/m^* embryos at 48 hpf. Each dot represents one embryo. Data are presented in mean ± SEM; one-way ANOVA with Dunnett's test, individual p-values mentioned in the figure. cryaa, αA-crystallin; PA, polyadenylation signal; SA, splice acceptor; T2A, T2A self-cleaving peptide. Individual scale bars are indicated in the figure.

*Figure 1 continued on next page*

*Figure 1 continued*

The online version of this article includes the following source data and figure supplement(s) for figure 1:

**Source data 1.** Original files for genomic PCR analysis displayed in *Figure 1B*.

**Source data 2.** File with bands labeled for genomic PCR analysis displayed in *Figure 1B*.

**Figure supplement 1.** Generation of *betaPix* conditional trap (*betaPix*[ct]) allele by an HDR-mediated knock-in method.

(*Figure 1—figure supplement 1C*), thus achieving ~1% efficiency on generating conditional alleles. Briefly, we established an efficient method for generating conditional knockin/knockout mutants in general and generated a conditional gene trap line Ki(*betaPix*:Zwitch) in the *betaPix* locus in particular, which is referred to as *betaPix*[ct] hereafter.

A previous work has shown that *bubblehead* (*bbh*[fn40a]) mutant has a global reduction in *betaPix* transcripts, and *bbh*[m292] mutant has a hypomorphic mutation in *betaPix*, thus establishing that *betaPix* is responsible for *bubblehead* mutant phenotypes (*Liu et al., 2007*). *bbh*[fn40a] mutants developed cerebral hemorrhages phenotype between 36 and 52 hpf (*Figure 1—figure supplement 1D and E*). To characterize the *betaPix*[ct] allele, we injected one-cell-stage *betaPix*[ct/ct] embryos with *Cre* mRNA to induce global inversion of Zwitch and deletion of *betaPix*. As expected, we found *Cre*-induced precise inversion in *betaPix*[m/m] as confirmed by Sanger sequencing (*Figure 1C*). By using o-dianisidine staining to label hemoglobins, we found severe brain hemorrhages developed in *betaPix*[m/m] mutant with the time window similar to *bbh*[fn40a] mutants (*Figure 1D and E*). Consistently, qPCR analysis showed that *betaPix* decreased and *RFP* increased in heterozygous *betaPix*[m/+] and homozygous *betaPix*[m/m] mutants compared with *betaPix*[ct/ct] wild-type siblings, while *alphaPix* expression was not affected (*Figure 1F*). Of note, *TagRFP* expression in *betaPix*[m/+] (after Cre-mediated recombination in *betaPix*[ct/+] embryos) reflected endogenous *betaPix* expression in the boundary of midbrains and hindbrains as well as the hindbrains by light-sheet fluorescence microscopy imaging (*Figure 1—figure supplement 1F and G*). Together, these results suggest that the conditional trap *betaPix*[ct/ct] zebrafish is functional for visualizing endogenous *betaPix* expression (knockin) and performing loss-of-function study (conditional knockout).

## *betaPix*[m/m] mutant has brain hemorrhages, central arteries defects, and abnormal glial structure that is partially rescued by Pak1 inhibitor IPA3 treatment

The main phenotypes in *bbh* mutants consist of brain hemorrhage and hydrocephalus as early as 36 hpf, as well as poor endothelial–mesenchymal contacts and defective central arteries (CtAs) sprouting in the hindbrain (*Liu et al., 2007*; *Liu et al., 2012*). To demonstrate the practicability of *betaPix* conditional trap lines, we assessed reported phenotypes associated with *betaPix* deficiency using homozygous *betaPix*[ct/ct] zebrafish. Global *betaPix* inactivation with *Cre* mRNA injection resulted in severe cerebral hemorrhages (*Figure 1D and E*). By crossing the *betaPix*[m/m] mutant line with Tg(*kdrl*:GFP) transgenic line, we found defective angiogenesis in hindbrain central arteries via light-sheet fluorescence microscopy imaging (*Figure 2—figure supplement 1A and B*), which is consistent with *betaPix* mutant phenotypes as reported (*Liu et al., 2007*; *Yang et al., 2017*).

It has been suggested that endothelial *betaPix* might not contribute to the occurrence of brain hemorrhage (*Liu et al., 2007*). During brain development, neurons and glia are important perivascular cells, and they orchestrate with endothelial cells in a temporal and spatial pattern. Transgenic zebrafish that express fluorescent proteins driven by the glial fibrillary acidic protein (*gfap*) promoter have been widely used for labeling glial population (*Bernardos and Raymond, 2006*). By crossing the *betaPix*[m/m] mutant line with Tg(*gfap*:GFP) transgenic line, we found that radial glial structure of the hindbrain had disoriented arrangement and shortened process length in *betaPix*[m/m] embryos compared to siblings by light-sheet fluorescence microscopy imaging (*Figure 2A and B*, *Figure 2—figure supplement 1C*). RNA in situ hybridization showed that neuronal and glial precursors marker *nestin* increased while differentiated neuronal marker *pax2a* decreased at the hindbrain of *betaPix*[m/m] embryos, suggesting delayed neuronal development and differentiation after *betaPix* inactivation (*Figure 2C*). By using global *betaPix* knockout with multiple guide-RNA simultaneously (*Wu et al., 2018*), we found that CRISPR-induced *betaPix* mutant $F_0$ embryos had almost identical phenotypes to those in *betaPix*[fn40a] mutants (*Figure 2—figure supplement 2A and B*), such as

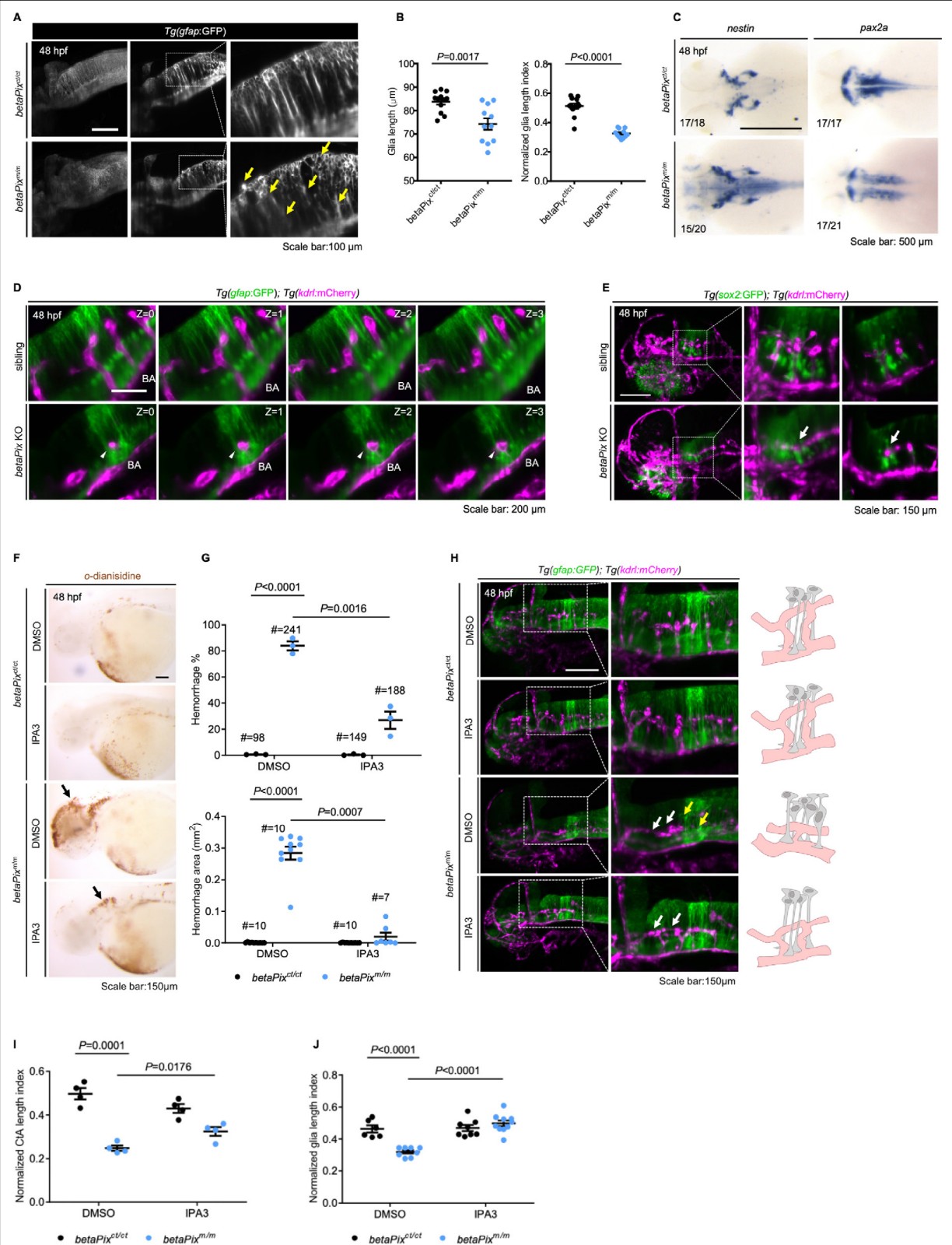

**Figure 2.** *betaPix^{m/m}* mutants have brain hemorrhages, central artery defects, and abnormal glial structure that was partially rescued by Pak1 inhibitor IPA-3 treatment. (**A**) Left panel showing the maximum intensity projection of the glial structures in the hindbrain of *betaPix^{ct/ct}* and *betaPix^{m/m}* embryos at 48 hpf. Lateral view, anterior to left. Middle panel showing representative optical sections and right panel showing the higher magnifications of boxed area, presenting atypical glial structures with disoriented arrangements (yellow arrows) in *betaPix^{m/m}* embryos. (**B**) Quantification of glial parameters

*Figure 2 continued on next page*

*Figure 2 continued*

in (**A**). Left panel showing the average glia length, and right panel showing glia length index normalized to individual head length, which each dot represents one embryo. Data are presented in mean ± SEM; unpaired Student's *t*-test with individual p-values mentioned in the figure. (**C**) Whole-mount RNA in situ hybridization revealed *nestin* and *pax2a* expression pattern in *betaPix*$^{ct/ct}$ and *betaPix*$^{m/m}$ embryos at 48 hpf. Dorsal view, anterior to the left. (**D**) Optical sections of glial structure (green) and blood vessels (magenta) in the heads of siblings and CRISPR-mediated *betaPix* F$_0$ knockout embryos. Arteries in the hindbrain of *betaPix* KO mutants had developmental defects (white arrowheads), showing shorter distance between basilar artery and glial cell bodies. (**E**) 3D reconstruction of the *sox2*-positive precursors (green) and vasculatures (magenta) in the heads of siblings and CRISPR-mediated *betaPix* F$_0$ knockout embryos at 48 hpf. Box areas are shown in higher magnifications at the middle panels, with optical sections shown in the right panels. Arrows indicate CtA with enlarged perivascular space. (**F**) Representative stereomicroscopy images of *o*-dianisidine staining of *betaPix*$^{ct/ct}$ and *betaPix*$^{m/m}$ embryos at 48 hpf that were treated with DMSO or PAK inhibitor IPA3. Brain hemorrhages indicated with arrows. (**G**) Quantification of brain hemorrhagic parameters in (**F**). Left panel showing hemorrhage percentages, with independent experiments as dots. Right panel showing hemorrhage areas, with each dot representing one embryo, # represents the numbers of embryos scored for each analysis, and three or more individual experiments conducted. Data are presented in mean ± SEM; one-way ANOVA with Dunnett's test, individual p-values mentioned in the figure. (**H**) Left panels showing 3D reconstruction of the glial structure (green) and vasculature (magenta) in the heads of *betaPix*$^{ct/ct}$ and *betaPix*$^{m/m}$ embryos at 48 hpf treated with DMSO or IPA3. Lateral view, anterior to left. Box areas are shown in higher magnifications at the middle panels. Defects in hindbrain central arteries are indicated in white arrows, while defects in radial glia are indicated in yellow arrows. Right panels showing schematic diagrams. Glia (gray) and CtAs (pink) develop normally in DMSO or IPA3-treated *betaPix*$^{ct/ct}$ embryos, with fine radial glial processes and characteristic arch vasculature. Yet in *betaPix*$^{m/m}$ embryos, abnormal development of the glia and central artery presented. In IPA-3-treated *betaPix*$^{m/m}$ embryos, central arterial defects were partially rescued with relatively complete arch architecture, and glial process defects were significantly rescued. (**I**) Quantification of CtA parameters in (**H**). Left panel showing the average CtA length, and right panel showing the CtA length index normalized to individual head length, with each dot representing one embryo. Data are presented in mean ± SEM; one-way ANOVA with Dunnett's test, individual p-values mentioned in the figure. (**J**) Quantification of glia parameters in (**H**). Left panel showing the average glia length, and right panel showing glia length index normalized to individual head length, which each dot represents one embryo. Individual scale bars are indicated in the figure. Data are presented in mean ± SEM; one-way ANOVA with Dunnett's test, individual p-values mentioned in the figure. Individual scale bars are indicated in the figure. BA, basilar artery.

The online version of this article includes the following figure supplement(s) for figure 2:

**Figure supplement 1.** *betaPix*$^{m/m}$ mutant had brain hemorrhages, central artery defects, and abnormal glial structure.

**Figure supplement 2.** CRISPR-mediated *betaPix* F$_0$ knockouts had similar phenotypes as *betaPix*$^{m/m}$ mutants.

cerebral hemorrhages, central artery defects, and abnormal hindbrain glial and neuronal precursor development (*Figure 2—figure supplement 2C–G*), further confirming the loss-of-function phenotypes in *betaPix*$^{m/m}$ mutants.

Vascularization of the central arteries in the zebrafish hindbrain starts at 29 hpf. Tip cells start to form in the dorsal side of the primordial hindbrain channels, sprout, and migrate into the hindbrain tissue, moving toward the center around 36 hpf to establish connections with the basilar artery. A characteristic arch architecture forms at 48 hpf as boundaries dividing the hindbrain into rhombomeres (*Quiñonez-Silvero et al., 2020*). In Tg(*gfap*:GFP);Tg(*kdrl*:mCherry) double transgenic embryos, CRISPR-induced *betaPix* knockouts showed that CtA sproutings were restricted to the bottom of glial processes that failed to migrate upwards to form an arched structure in comparison with non-injected siblings (*Figure 2D*). The transcription factor SRY box-2 (Sox2) is another marker gene for neuron and glial precursors. In Tg(*sox2*:GFP); Tg(*kdrl*:mCherry) transgenic background, *Sox2*-positive precursor cells tightly wrapped around the central artery in control siblings, but *Sox2* precursor cells had loose contacts with the central arteries, with larger perivascular distances in CRISPR-induced *betaPix* mutants (*Figure 2E*). These results implicate the impaired interactions between endothelial cells and neuronal/glial cells during development after *betaPix* deletion.

p21-activated kinase (Pak) is a binding partner for *betaPix*. *Pak2a* has been shown to mediate downstream signaling in *bbh*$^{m292}$ mutants. Group I PAK allosteric inhibitor IPA-3 covalently modifies and stabilizes the autoinhibitory N-terminal region of PAK1, PAK2, and PAK3 (*Deacon et al., 2008*). As expected, IPA-3 treatment significantly decreased incidence and intensity of cerebral hemorrhages in *betaPix*$^{m/m}$ mutant embryos (*Figure 2F and G*), while IPA-3 treatment had no effect on hemorrhage induction in *betaPix*$^{ct/ct}$ control siblings. In addition, central arterial defects were partially rescued in *betaPix*$^{m/m}$ mutant embryos after IPA-3 treatment, showing increased endothelial protruding into the hindbrain and more arch structure formation (*Figure 2H and I*). IPA-3 treatment also decreased abnormal glial process arrangements, which the lengths of glial processes statistically reached the levels of the control siblings (*Figure 2J*). Thus, these data suggest that the *betaPix-Pak1/2* signaling regulates brain vascular integrity during development.

## Glial-specific *betaPix* knockouts recapture its global knockout phenotypes

To investigate *betaPix* function in different types of perivascular cells, we generated glial- or neuronal-specific transgenic lines using either *gfap* (*Bernardos and Raymond, 2006*) or *huC* (*Kim et al., 1996*) promoters to drive *EGFP-Cre* fusion gene expression under *betaPix$^{ct/ct}$* background, respectively (*Figure 3—figure supplement 1A*). EGFP signals reported *Cre* expression while RFP signals efficiently reported *Cre*-induced gene trap cassette inversion and disruption of *betaPix* expressions (*Figure 3— figure supplement 1B–F*). Interestingly, glial-specific *betaPix* knockouts developed severe cerebral hemorrhages and abnormal central arteries vascularization, which were partially rescued by IPA-3 treatment (*Figure 3A–C*), highlighting *betaPix* function in glia via Pak1/2 signaling. On the other hand, neither vascular-, neuronal-, nor mural-specific deletion of *betaPix* had evident *betaPix* mutant phenotypes (*Figure 3—figure supplement 2A–D*). These results suggest that glial *betaPix* plays crucial roles in zebrafish embryonic vascular integrity and glial development via regulating *Pak* activities.

## Single-cell transcriptome profiling reveals that *gfap*-positive progenitors were affected in *betaPix* knockouts

To investigate the interplays between glial and hemorrhagic pathology caused by *betaPix* loss-of-function, we profiled the cranial tissues of CRISPR-edited zebrafish at 1 and 2 dpf using the 10X Chromium single-cell RNA sequencing (scRNA-seq) platform (*Figure 4A*). Multi-guide targeting was able to generate almost 100% $F_0$ null mutants, allowing us to select mutant embryos before brain hemorrhages starting from 36 hpf. Low-quality cells were excluded based on the numbers of genes detected and percentages of reads mapped to mitochondrial genes per sample. A total of 38,670 cells passed quality control and were used for subsequent analyses. Uniform manifold approximation and projection (UMAP) identified 71 cell clusters, which represented 24 zebrafish cranial cell types based on known marker gene expression profiles (*Figure 4B*, *Figure 4—figure supplement 1A*). Enriched gene markers were compared with previously annotated gene markers in the ZFIN database and literatures (*Raj et al., 2018*; *Raj et al., 2020*). By comparing the proportion of cells in each sample, we found that most neuronal clusters had increased relative proportions, while the numbers of glial and neuronal progenitors, endothelial cells, erythrocytes, neural crests, muscles, cartilages, retinas (photoreceptor precursor cells), olfactory bulbs, or epidermis and pharyngeal arches were reduced in *betaPix* knockout heads at 2 dpf (*Figure 4C*, *Figure 4—figure supplement 1B*). This is consistent with the open public databases of single-cell transcriptome atlas where *betaPix* is weakly expressed in a broad range of cells including glia, neurons and neuronal precursors, retinas, endothelial cells, pharynx, exocrine pancreas, olfactory cells, and heart cells (*Farnsworth et al., 2020*). Given that *betaPix* is critical in glia during brain vascular integrity development as shown above (*Figures 1 and 2*), we examined the *gfap* expression among each cluster and found relatively high *gfap* expression in clusters including glial and neuronal progenitors, hindbrain, ventral diencephalon, ventral midbrain, and floor plate (*Figure 4—figure supplement 1A*, indicated by the arrow). We next focused on the progenitor cluster owing to the enriched *gfap* expression and the significantly reduced numbers of cells in this cluster by *betaPix* deficiency. Furthermore, this progenitor cluster exhibited high-level expression of cell proliferation and cell cycle-related genes (*mki67*, *pcna*, *ccnd1*, *rrm1*, and *rrm2*) as well as key glial-associated genes (*gfap*, *fabp7a*, *her4*.1, *cx43*, *id1*, *fgfbp3*, *atp1a1b*, and *mdka*) (*Supplementary file 1*). From differentially expressed genes in this progenitor cluster between controls and *betaPix* knockouts at 2 dpf, gene ontology (GO) terms revealed three major categories: epigenetic remodeling, microtubule organizations, and neurotransmitter secretion/transportation (*Figure 4D*). Of these signaling genes, we were particularly interested in microtubule organizing genes for further studies.

## Stathmin acts downstream of *betaPix* in glial migration via regulating tubulin polymerization

Microtubules are essential cytoskeletal elements composed of α/β-tubulin heterodimers. The regulation of microtubule polymerization affects important cellular functions such as mitosis, motor transport, and migration (*Etienne-Manneville, 2013*). Based on the above scRNA-seq data (*Figure 4E*, *Figure 4—figure supplement 1C*), we examined if microtubule-destabilizing protein Stathmins were affected in *betaPix* mutants. We found that *stathmin1a* (*stmn1a*), *stathmin1b* (*stmn1b*), and *stathmin4l* (*stmn4l*) were relatively abundant in both the whole brain and the progenitor cluster and decreased

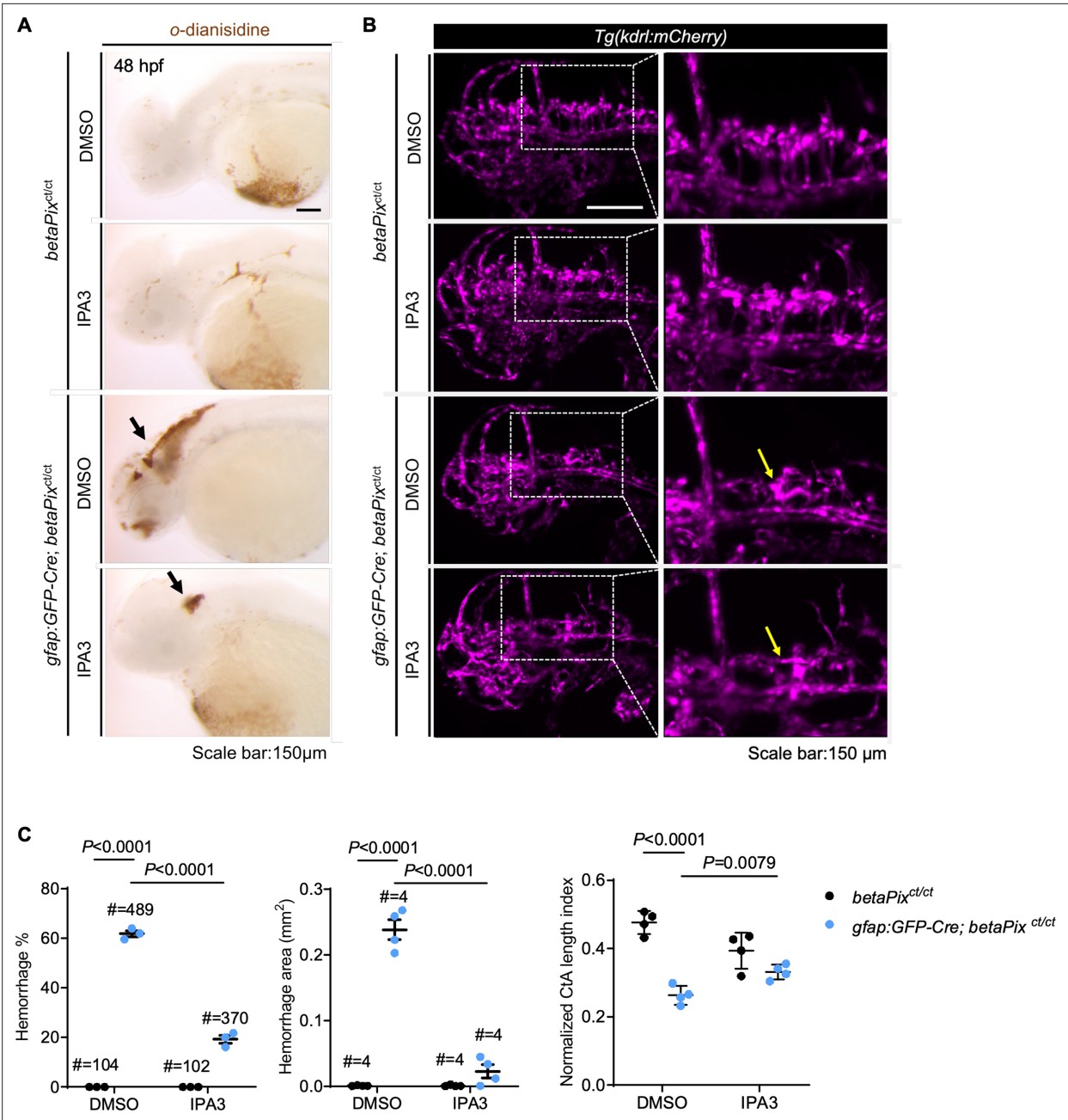

**Figure 3.** Glial-specific *betaPix* knockouts recapture global *betaPix* mutant phenotypes. (**A**) Representative stereomicroscopy images of erythrocytes stained with *o*-dianisidine in *betaPix*^ct/ct^ siblings and *gfap:GFP-Cre; betaPix*^ct/ct^ mutant embryos treated with DMSO or IPA3 at 48 hpf. Brain hemorrhages indicated with arrows in glial-specific *betaPix* knockouts. Lateral view with anterior to the left. (**B**) Left panels showing 3D reconstruction of the vasculature (magenta) in the heads at 48 hpf, lateral view with anterior to the left. Box areas are shown in higher magnifications of brain vasculatures at the right panels. CtA defects indicated in yellow arrows in *gfap:GFP-Cre; betaPix*^ct/ct^ mutant embryos. (**C**) Quantification of brain hemorrhages in (**A**) and CtA parameters in (**B**). Left panel showing hemorrhage percentages, with independent experiments as dots. Middle panel showing hemorrhage areas with each dot representing one embryo. # represents the numbers of embryos scored for each analysis, three or more individual experiments conducted. Right panel showing CtA length index normalized to individual head length, with each dot representing one embryo. Data are presented in mean ± SEM; one-way ANOVA with Dunnett's test, individual p-values mentioned in the figure. Individual scale bars are indicated in the figure.

The online version of this article includes the following source data and figure supplement(s) for figure 3:

**Figure supplement 1.** Generating glial-specific *betaPix* knockout zebrafish.

**Figure supplement 1—source data 1.** Original files for genomic PCR analysis displayed in *Figure 3—figure supplement 1C*.

**Figure supplement 1—source data 2.** File with bands labeled for genomic PCR analysis displayed in *Figure 3—figure supplement 1C*.

*Figure 3 continued on next page*

*Figure 3 continued*

**Figure supplement 2.** Neither endothelial-, neuronal-, nor mural-specific deletion of *betaPix* caused brain hemorrhages and abnormal CtA phenotypes.

specifically at 2 dpf in *betaPix* knockout progenitor cells (*Figure 4E*). Subsequent qRT-PCR revealed that these *Stathmin* expression decreased in both global *betaPix* knockouts (*betaPix^{m/m}*) and glial-specific *betaPix* knockout glia (*gfap*:EGFP-Cre;*betaPix^{ct/ct}*) (*Figure 4F and G*), which was further verified by RNA in situ hybridization analysis (*Figure 4H*). We next investigated whether *Stathmins* act downstream to *betaPix* signaling on regulating vascular integrity development. To this end, we found that *Pak1/2* inhibition enabled the rescue of downregulated expressions of *stmn1a*, *1b*, and *4l* in either global *betaPix^{m/m}* or glial-specific *betaPix* knockout zebrafish (*Figure 5A and B*, *Figure 5— figure supplement 1A*), which is consistent with previous findings that *Stathmin-1* acts downstream to *betaPix-Pak* signaling in neurite outgrowth in mice (*Kwon et al., 2020*).

To test whether *Stathmin* family genes are important for vascular stability, we utilized CRISPR-induced $F_0$ knockout system simultaneously targeting the three stathmin genes. While singular stathmin gene knockout led to no evident phenotypes (data not shown), we found that simultaneously inactivating *stmn1a*, *stmn1b*, and *stmn4l* caused brain hemorrhages and hydrocephalus in the brain, similar to that from *betaPix^{fn40a}* and *betaPix^{m/m}* mutants (*Figure 5C and D*, *Figure 5—figure supplement 1B–E*). By labeling both glia and vasculature with Tg(*gfap*:GFP; *kdrl*:mCherrry) transgenic embryos, we found that triple stathmin gene knockouts had impaired CtAs, abnormal glia and neuronal development in the hindbrain region (*Figure 5E and F*, *Figure 5—figure supplement 1F–H*). Furthermore, overexpressing *stmn1b* by the *gfap* promoter partially rescued brain hemorrhages and delayed neuronal development in *bbh^{fn40a}* mutants (*Figure 5G and H*, *Figure 5—figure supplement 1I and J*), of which glial processes elongation also improved, but CtAs defects failed to restore (*Figure 5I and J*, *Figure 5—figure supplement 1K*). Thus, these data support the critical role of *stathmins* in glia.

To examine the direct role of *betaPix-stathmin* in glia, we turned to use human glioblastoma cell lines and manipulated the *betaPIX* levels by siRNA knockdown or transgenic overexpression. We found that *betaPIX* knockdown attenuated glial migration (*Figure 5K and L*, *Figure 5—figure supplement 1L–N*), which is consistent with previous studies on *betaPix* function in the endoderm (*Omelchenko et al., 2014*), epithelial cells (*Hsu et al., 2014*), neurons (*López Tobón et al., 2018*), and multiple carcinoma cell lines including glioblastoma (*Cheng et al., 2023*; *Connor et al., 2020*; *Ward et al., 2015*). Either *betaPIX* or *STMN1* overexpression led to a partial restoration of the impaired glial migration, suggesting *Stathmins* as downstream effectors of *betaPix*. Furthermore, immunofluorescence analysis revealed that *betaPIX* knockdown led to abnormal accumulation of microtubules at the cell periphery, whereas ectopic expression of *betaPIX* and *Stathmin* partially rescued these microtubule defects (*Figure 5M and N*). Together, these results suggest that *betaPix* depletion disrupts microtubule dynamics and leads to impaired glial development acting upstream to the *Pak-Stathmin* signaling, subsequently disrupting vascular integrity and resulting in brain hemorrhages. While the brain hemorrhages were only partially rescued and abnormal CtA defects failed to restore, indicating that additional pathways may act downstream of *betaPix*.

### *Zfhx3/4* acts downstream of *betaPix* in regulating vascular integrity development

Previous studies have identified multiple signaling pathways important for zebrafish hindbrain angiogenic sprouting, such as VEGF, Notch, Wnt, Integrin β1, angiopoietin/TIE, and insulin-like growth factor signaling (*Bussmann et al., 2011*; *Eubelen et al., 2018*; *Pitulescu and Adams, 2014*; *Wang et al., 2021*). To investigate whether angiogenic signal is disrupted by *betaPix* depletion, we examined several angiogenic gene expression patterns by performing RNA in situ hybridization of embryos at 36 hpf. The expression level of *Vegfaa* in sprouting CtAs decreased in *bbh^{fn40a}* mutants (*Figure 6A*). Consistently, *betaPix* knockdown significantly reduced *VEGFA* expression in cultured glial cells, which was rescued by ectopic expression of *betaPix* (*Figure 6B*; right panel). To explore how *betaPix* regulate *Vegf* signaling in glia, we re-examined the GO analysis of single-cell sequencing data and found a significant enrichment of epigenetic and transcriptional regulation in glial progenitor cluster (*Figure 4D*). Among these candidate genes, transcription factor Zinc Finger Homeobox 3 (ZFHX3) is known to mediate hypoxia-induced angiogenesis in hepatocellular carcinoma *via* transcriptional

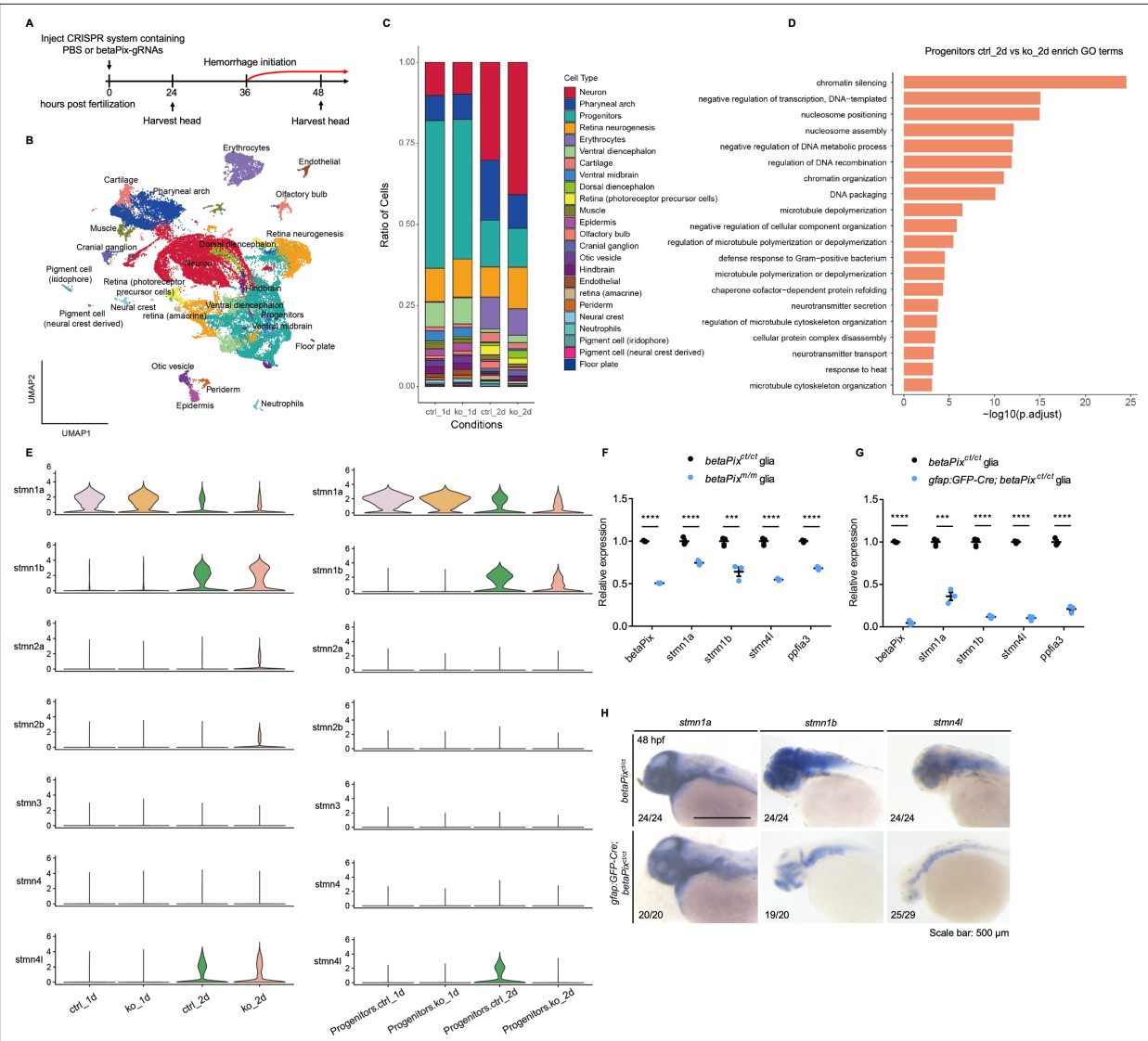

**Figure 4.** Single-cell transcriptome reveals that a subcluster of glial progenitor and stathmin family members are associated with *betaPix* mutation. (**A**) Experimental strategy for single-cell RNA sequencing of embryonic heads from wild-type siblings and *betaPix* CRISPR mutants at 1 dpf and 2 dpf. (**B**) UMAP visualization and clustering of cells labeled by cell type. Four samples were aggregated and analyzed together. (**C**) Proportions of 24 cell clusters were differentially distributed among four sample groups. ctrl_1d, PBS-injected siblings at 1 dpf; ko_1d, *betaPix* CRISPR mutants at 1 dpf; ctrl_2d, PBS-injected siblings at 2 dpf; ko_2d, *betaPix* CRISPR mutants at 2 dpf. (**D**) Enriched GO terms for differentially expressed genes for progenitor sub-cluster comparing ko_2d to ctrl_2d groups. (**E**) Violin plots showing the expression of the stathmin family genes by all cells among four sample groups (left panel) or by progenitor sub-cluster among four sample groups (right panel). (**F**) qRT-PCR analysis showing expression of *betaPix*, *stmn1a*, *stmn1b*, *stmn4l*, and *ppfia3* in glia of FACS-sorted *betaPix*$^{ct/ct}$ siblings and *betaPix*$^{m/m}$ mutants at 48 hpf. Each dot represents cells sorted from one embryo. Data are presented in mean ± SEM. ***p<0.005; ****p<0.001; unpaired Student's *t*-test. (**G**) qRT-PCR analysis showing expression of *betaPix*, *stmn1a*, *stmn1b*, *stmn4l*, and *ppfia3* in glia of FACS-sorted *betaPix*$^{ct/ct}$ siblings and *gfap:GFP-Cre; betaPix*$^{ct/ct}$ mutants at 48 hpf. Each dot represents cells sorted from one embryo. Data are presented in mean ± SEM. ***p<0.005; ****p<0.001; unpaired Student's *t*-test. (**H**) Whole-mount RNA in situ hybridization revealing downregulation of *stmn1a*, *stmn1b*, and *stmn4l* in *betaPix*$^{ct/ct}$ siblings and *gfap:GFP-Cre; betaPix*$^{ct/ct}$ mutant embryos at 48 hpf. Individual scale bars are indicated in the figure.

The online version of this article includes the following figure supplement(s) for figure 4:

**Figure supplement 1.** Glial progenitor sub-cluster of *betaPix* knockouts has downregulation of *stathmin* family members and up-regulation of *pak1* gene.

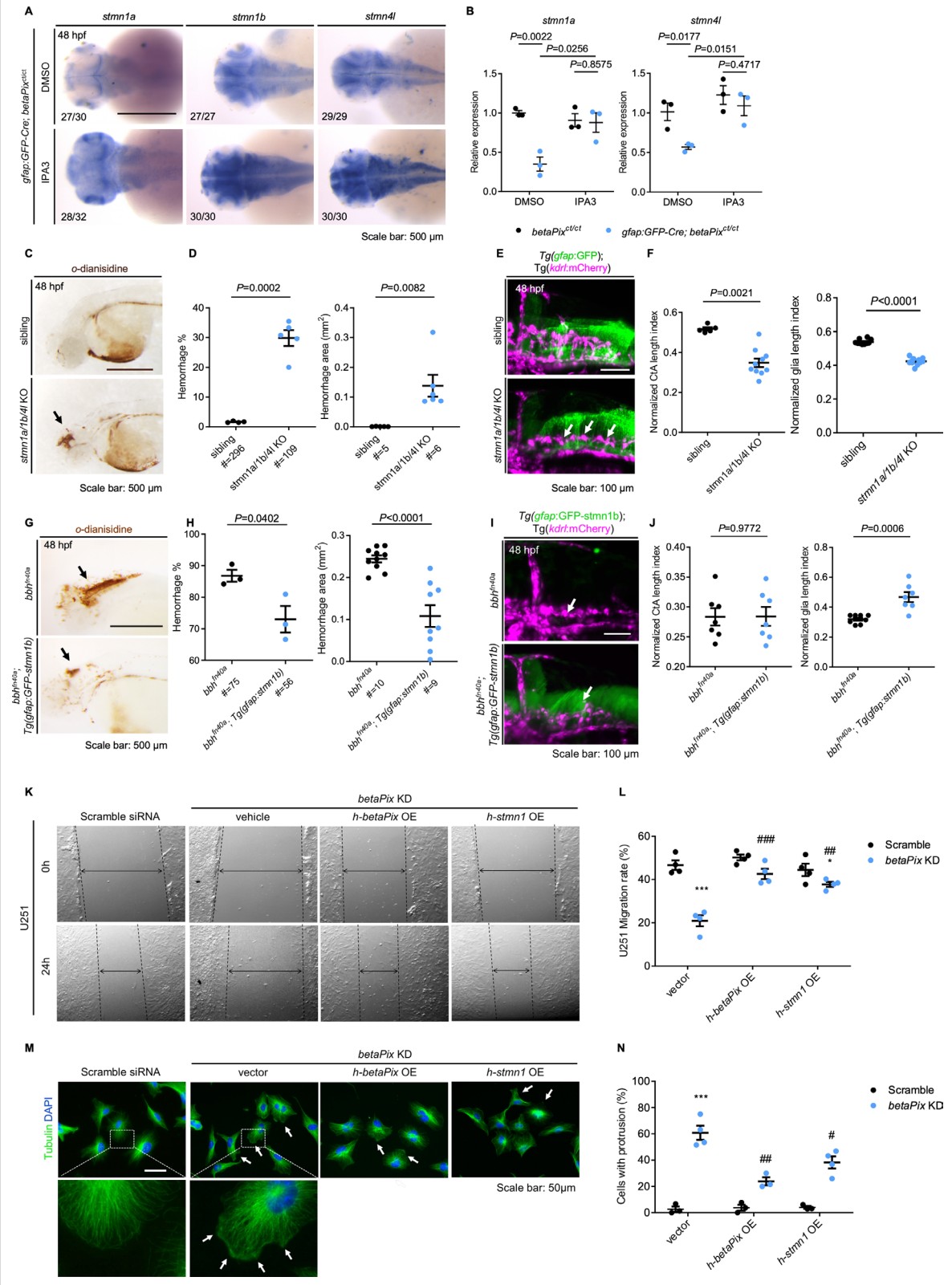

**Figure 5.** *Stathmin* acts downstream of *betaPix* in glial migration via regulating tubulin polymerization. (**A**) Whole-mount RNA in situ hybridization showing that *stmn1a, stmn1b, and stmn4l* expression in *gfap:GFP-Cre; betaPix^ct/ct* embryos were partially rescued by Pak1 inhibitor IPA3 treatment at 48 hpf. Dorsal views, anterior to the left. (**B**) qRT-PCR analysis showing that *stmn1a* and *stmn4l* expression were rescued in *gfap:GFP-Cre; betaPix^ct/ct* mutants by IPA3 treatment at 48 hpf. Each dot represents one embryo. Data are presented in mean ± SEM; one-way ANOVA with Dunnett's test,

*Figure 5 continued on next page*

*Figure 5 continued*

individual p-values mentioned in the figure. (**C**) Representative stereomicroscopy images of erythrocytes stained with *o*-dianisidine in siblings and *stmn1a/1b/4l* CRISPR mutants at 48 hpf. Brain hemorrhages, indicated with arrows, appeared in *stmn1a/1b/4l* mutants. Lateral views with anterior to the left. (**D**) Quantification of hemorrhagic parameters in (**C**). Left panel showing hemorrhage percentages, with independent experiment as dot. Right panel showing hemorrhage areas with each dot representing one embryo. # represents the numbers of embryos scored for each analysis, three or more individual experiments conducted. Data are presented in mean ± SEM; unpaired Student's *t*-test with individual p-values mentioned in the figure. (**E**) 3D reconstruction of glial structure (green) and vasculature (magenta) in the heads at 48 hpf. Lateral view with anterior to the left. CtA defects (white arrows) indicated in *stmn1a/1b/4l* mutants. (**F**) Quantification of CtA and glia parameters in (**E**). Length index normalized to individual head length, with each dot representing one embryo. Data are presented in mean ± SEM; unpaired Student's *t*-test with individual p-values mentioned in the figure. (**G**) Representative stereomicroscopy images of erythrocytes stained with *o*-dianisidine in *bbh^fn40a^* and *bbh^fn40a^*; Tg(*gfap:GFP-stmn1b*) embryos at 48 hpf. Brain hemorrhages, indicated with arrows, decreased in *bbh^fn40a^* mutants with glia-specific overexpression of *stmn1b*, compared with *bbh^fn40a^* mutant siblings. Lateral views with anterior to the left. (**H**) Quantification of hemorrhagic parameters of (**G**). Left panel showing hemorrhage percentages, with independent experiment as dot. Right panel showing hemorrhage areas with each dot representing one embryo. # represents the numbers of embryos scored for each analysis, three or more individual experiments conducted. Data are presented in mean ± SEM; unpaired Student's *t*-test with individual p-values mentioned in the figure. (**I**) 3D reconstruction of the *gfap:GFP-stmn1b* overexpression (green) and vasculature (magenta) in the heads of *bbh^fn40a^* siblings and *bbh^fn40a^*; Tg(*gfap:GFP-stmn1b*) mutants at 48 hpf. Lateral view with anterior to left. White arrows indicate CtA defects. (**J**) Quantification of CtA and glia parameters in (**I**). Length index normalized to individual head length, with each dot representing one embryo. Data are presented in mean ± SEM; unpaired Student's *t*-test with individual p-values mentioned in the figure. (**K**) Representative stereomicroscopy images of U251 cells at 0 and 24 hours after wounding. U251 cells were transfected with negative control siRNA or *betaPIX* siRNA separately, in combination with pcDNA3.1 vector, *betaPIX* overexpression plasmid, and *STMN1* overexpression plasmid. The wound edges are highlighted by dashed lines, with arrow lines indicating the wound width. (**L**) Quantification of wound closure in (**K**), showing *p<0.05, ***p<0.005 compared to negative control siRNA with empty vector transfection. ##p<0.01, ###p<0.005 compared to *betaPix* knockdown with empty vector transfection. Data are presented in mean ± SEM; one-way ANOVA with Dunnett's test. (**M**) Representative immunofluorescence image of alpha-tubulin and DAPI signals in U251 cells. U251 cells were transfected with negative control siRNA or *betaPIX* siRNA separately, in combination with pcDNA3.1 vector, *betaPIX* overexpression plasmid, and *STMN1* overexpression plasmid. Box areas are shown in higher magnifications. Arrows indicate protrusions at the cell periphery. (**N**) Quantification of cell percentages with protrusions in (**M**). ***p<0.005 compared to negative control siRNA with empty vector transfection. #p<0.05, ##p<0.01 compared to *betaPix* knockdown with empty vector transfection. Data are presented in mean ± SEM; one-way ANOVA with Dunnett's test. Individual scale bars are indicated in the figure.

The online version of this article includes the following figure supplement(s) for figure 5:

**Figure supplement 1.** CRISPR-mediated *stmn1a/1b/4* l ~F0~ knockouts have similar but milder phenotypes as *betaPix* knockouts.

---

activation of VEGFA (**Fu et al., 2020**). Furthermore, ZFHX3 has been associated with stroke in multiple genome-wide association studies (**Gudbjartsson et al., 2009**; **Soriano-Tárraga et al., 2020**; **Malik et al., 2016**). Interestingly, we found that transcription factors *ZFHX3* and *ZFHX4* significantly down-regulated after *betaPIX* inactivation in cultured glial cells (**Figure 6B**). The transcription levels of *Zfhx3* and *Zfhx4* also downregulated in the glial progenitor cluster of our single-cell RNA sequencing data upon *betaPix* knockout at 2 dpf (**Figure 6—figure supplement 1A**), as well as the glia-specific *betaPix* mutants (**Figure 6—figure supplement 1B**), thus suggesting that Zfhx3/4 might act as downstream effectors of *betaPix*.

To determine whether *Zfhx3/4* is critical for vascular integrity by interacting with *betaPix* in zebrafish, we depleted *Zfhx3* and *Zfhx4* simultaneously *via* CRISPR-mediated F~0~ knockouts (**Figure 6—figure supplement 1C–E**). We found that *Zfhx3/4* knockouts decreased *Vegfaa* expression at sprouting CtAs at 36 hpf (**Figure 6C**). *Zfhx3/4* knockouts at 48 hpf had under-developed CtAs and reduced penetration of endothelial cells into the hindbrain (**Figure 6F and G**), developed cerebral hemorrhages during 36–52 hpf, which are comparable with *betaPix* mutant phenotypes (**Figure 6D–G**). Despite upregulated expression level of *nestin*, glial arrangement and processes elongation remained statistically unaffected after *Zfhx3/4* inactivation (**Figure 6F and G**, **Figure 6—figure supplement 1F**). We next investigated whether increasing expression level of *Vegfaa* in *betaPix* mutants was able to rescue impaired vascular integrity. By using *Zfhx4* mRNA microinjection into one-cell stage embryos in either *bbh^fn40a^* mutants or CRISPR-induced *betaPix* knockouts, we found that ectopic expression of *Zfhx4* increased *Vegfaa* expression level (**Figure 6L**), thus leading to a drastic reduction of both percentages and volumes of brain hemorrhage shown in these global *betaPix* mutants (**Figure 6H and I**, **Figure 6—figure supplement 1G–I**), as well as improved CtAs and glial development (**Figure 6J and K**). To further explore whether angiogenic potential altered in the endothelial cells, we re-examined our scRNA-seq data and noted significant reduction of endothelial proportions among *betaPix* knockouts (**Figure 4—figure supplement 1B**), which showed similar trend as glial progenitors. In this endothelial cluster, CRISPR-induced *betaPix* knockouts had downregulated angiogenic gene expression at 2 dpf (**Figure 6M**). Despite the lack of endothelial phenotype in cellular resolution in our present

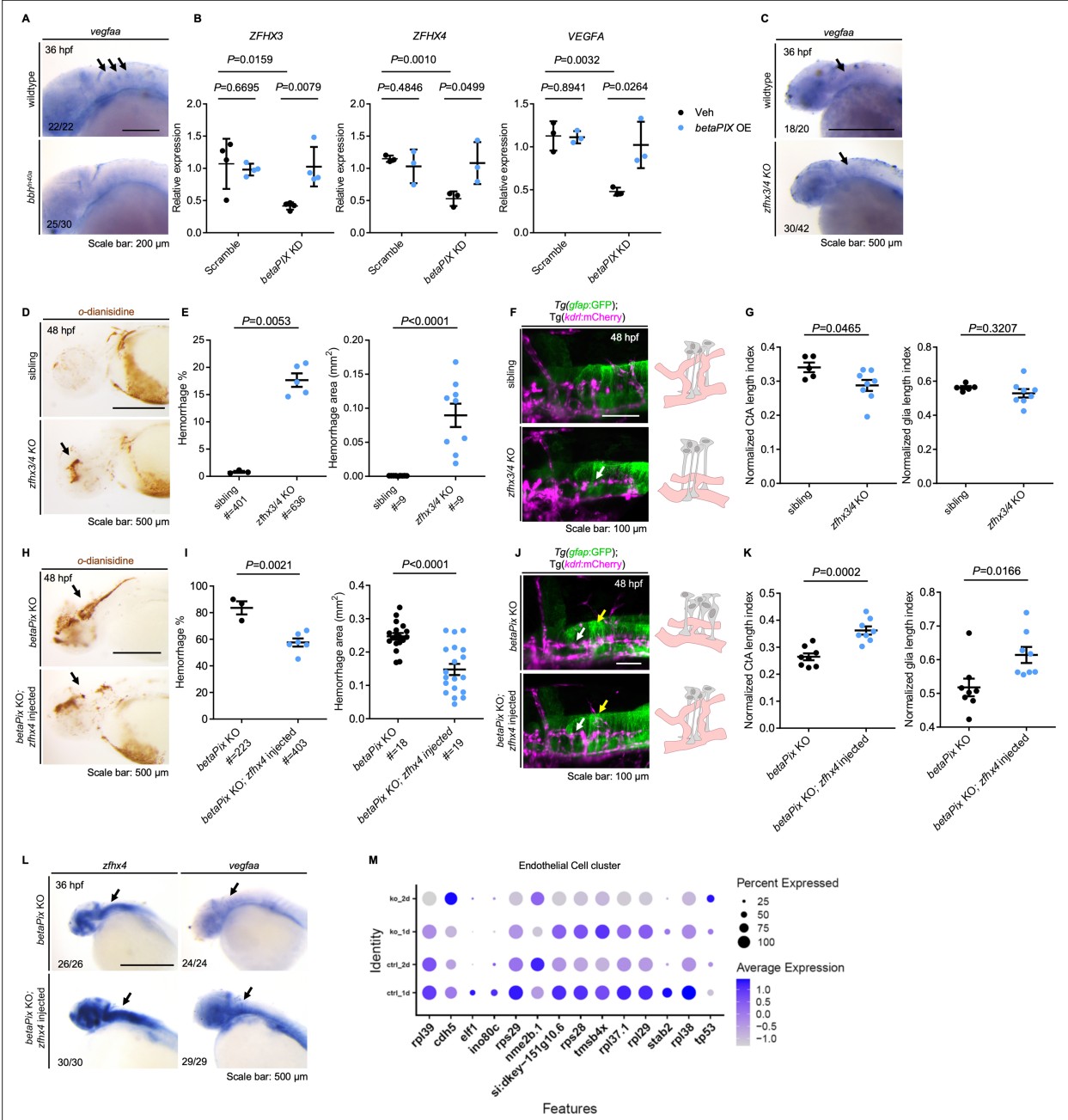

**Figure 6.** *Zfhx3/4* acts downstream of *betaPix* to regulate vascular integrity development. (**A**) Whole-mount RNA in situ hybridization revealing that *Vegfaa* decreased in *bbh*[fn40a] mutants compared with siblings at 36 hpf. Lateral views with anterior to the left. Arrows indicate *vegfaa* expression in the CtAs that showed depletion in *bbh*[fn40a] mutants. (**B**) qRT-PCR analysis revealing that *ZFHX3, ZFHX4,* and VEGFA decreased in U251 cells transfected with *betaPIX* siRNA, which were rescued by *betaPIX* overexpression. Data are presented in mean ± SEM; one-way ANOVA with Dunnett's test, individual p-values mentioned in the figure. (**C**) Whole-mount RNA in situ hybridization showing that *vegfaa* decreased in CRISPR-mediated *zfhx3/4* $F_0$ knockout embryos at 36 hpf. Lateral views with anterior to the left. Arrows indicate *vegfaa* expression in the CtAs that showed reduction in *zfhx3/4* knockouts. (**D**) Representative stereomicroscopy images of erythrocytes stained with *o*-dianisidine in siblings and CRISPR-mediated *zfhx3/4* $F_0$ knockout embryos at 48 hpf. Arrows indicated brain hemorrhages in the knockout brains. Lateral views with anterior to the left. (**E**) Quantification of hemorrhagic parameters in (**D**). Left panel showing hemorrhage percentages, with independent experiment as dot. Right panel showing hemorrhage areas with each dot representing one embryo. # represents the numbers of embryos scored for each analysis, three or more individual experiments conducted. Data are presented in mean ± SEM; unpaired Student's *t*-test with individual p-values mentioned in the figure. (**F**) Left panel showing 3D reconstruction of the glial structure (green) and vasculature (magenta) in the hindbrain of siblings and CRISPR-mediated *zfhx3/4* $F_0$ knockouts at 48 hpf. Lateral view with anterior to the left. White arrows indicate CtA defects. Right panels show schematic diagrams. Glia (gray) developed normally in both treatments, while defects in central artery (pink) presented in *zfhx3/4* knockout embryos. (**G**) Quantification of CtA and glia length parameters of (**F**). Length index

*Figure 6 continued on next page*

*Figure 6 continued*

normalized to individual head length, with each dot representing one embryo. Data are presented in mean ± SEM; unpaired Student's *t*-test with individual p-values mentioned in the figure. (**H**) Representative stereomicroscopy images of erythrocytes stained with *o*-dianisidine in CRISPR-mediated *betaPix* F0 knockout embryos with or without *zfhx4* mRNA injection at 48 hpf. Arrows indicate brain hemorrhages. Lateral views with anterior to the left. (**I**) Quantification of hemorrhagic parameters in (**H**). Left panel showing hemorrhage percentages, with independent experiment as dot. Right panel showing hemorrhage areas with each dot representing one embryo. # represents the numbers of embryos scored for each analysis, three or more individual experiments conducted. Data are presented in mean ± SEM; unpaired Student's *t*-test with individual p-values mentioned in the figure. (**J**) Left panel showing 3D reconstruction of the glial structure (green) and vasculature (magenta) in CRISPR-mediated *betaPix* $F_0$ knockout embryos with or without *zfhx4* mRNA injection at 48 hpf. White arrows indicate CtAs and yellow arrows indicate glia. Right panels show schematic diagrams. Glia (gray) and CtA (pink) developmental defects rescued in *zfhx4* treatment. (**K**) Quantification of CtA and glia length parameters in (**J**). Length index normalized to individual head length, with each dot representing one embryo. Data are presented in mean ± SEM; unpaired Student's *t*-test with individual p-values mentioned in the figure. (**L**) Whole-mount RNA in situ hybridization revealed that *Zfhx4* and *Vegfaa* decreased in CRISPR-mediated *betaPix* $F_0$ knockout embryos at 36 hpf, which were rescued by *Zfhx4* mRNA injection. Lateral views with anterior to the left. Arrows indicate hindbrain regions. (**M**) Dot plots of several angiogenesis-associated genes expression in endothelial cell cluster. Dot size indicates the percentage of cells with gene expression, and dot color represents the average gene expression level. Individual scale bars are indicated in the figure.

The online version of this article includes the following figure supplement(s) for figure 6:

**Figure supplement 1.** *Zfhx3/4* acts downstream of *betaPix* to regulate vascular integrity development.

results, the disruptive transcriptome signature in the endothelial cluster suggests a potential link to the defective central arterial development. Together, these data support the critical role of *Zfhx3/4* in vascular integrity and glial development, probably acting upstream to VEGFA and downstream to *betaPix*.

## Discussion

Conditional knockout technology in mice has been widely used to modify and determine targeted gene function in a cell-specific manner. Spatio-temporal specific modification is beneficial for studying embryonic lethality genes or investigating the function of targeted genes in specific cell populations. Despite being broadly utilized in mice, conditional knockouts in other animal models are limited due to technical difficulties. To this end, multiple research groups have designed different strategies for establishing floxed-mutant alleles, gene traps, and inducible Cas9 in zebrafish (*Kalvaityté and Balciunas, 2022*). In particular, the Zwitch gene trap-based approach has been established successfully in zebrafish, enabling to simultaneously produce knockin reporter driven by the endogenous gene promoter and disrupting endogenous gene function in specific cell types (*Sugimoto et al., 2017*; *Ogawa et al., 2021*; *Ogawa and Kikuchi, 2024*). We adapted this strategy with the aid of CRISPR/Cas9 technology and added a GSG spacer to enhance cleavage efficiency and universal guide RNA target sites on both homologous arms for producing linearized targeting vector. Moreover, we also found that utilizing either short or long homologous arms enables us to achieve precise targeted integrations with this donor vector. Our short arms-based method can be broadly adapted because of fewer restrictions on arm sequences and easily engineering the donor vector. From the *betaPix* locus and other five unpublished loci that we already generated, we achieved Zwitch alleles with ~1% correct insertions based on homologous recombination as well as *Cre*-induced, cell-specific deletion and functional analysis of targeted genes in both embryos and adult hearts. Therefore, our CRISPR-based Zwitch method can be widely applied for generating conditional mutant zebrafish in particular and possibly extended for other animal mutants in general.

The maturation hallmark of central nervous system (CNS) vasculature is acquisition of blood–brain barrier (BBB) properties, establishing a stable environment crucial for brain homeostasis in response to extrinsic factors and physiological changes. The core features of the BBB include specialized tight junctions, highly selective transporters, and limited immune cell trafficking (*Obermeier et al., 2013*; *Langen et al., 2019*). In zebrafish, BBB can be functionally characterized from 3 dpf (*Fleming et al., 2013*; *Xie et al., 2010*). *Bubblehead* mutants acquire vessel rupture phenotype early before BBB maturation, presenting a good hemorrhagic model for studying early BBB development in this work. Multiple types of cells contribute to diverse aspects of BBB development and maintenance, leading to proper vasculature function across developmental timelines. Endothelial cells form the continuous inner layer of blood vessels with junctional proteins and selective transporters, which are the main subject of permeability regulation in BBB. It is well accepted that endothelial cells do not show a

predetermined role and that the brain environment is sufficient to induce endothelial barrier (*Stewart and Wiley, 1981*). Nevertheless, accumulative evidence points to the critical role of endothelial angiocrine signals in regulating several aspects of BBB development. For example, Wnt7a/b ligands secreted by neural progenitor cells bind to Frizzled receptors on endothelial cells, activating canonical Wnt signaling and downstream genes. WNT ligands knockouts lead to angiogenic defects and BBB leakages specifically in the CNS (*Daneman et al., 2009*). Endothelial GPR124, as a coactivator of canonical Wnt signaling, has important function in CNS-specific angiogenesis and BBB establishment (*Kuhnert et al., 2010*; *Cullen et al., 2011*; *Zhou and Nathans, 2014*). Other signaling ligands such as angiopoietins and Ephrin family ligands orchestrate with Tie or EphB receptors on endothelial cells, respectively (*Thomas and Augustin, 2009*).

Complex interactions between BBB cell types orchestrate proper angiogenesis and CNS development in a spatio-temporal manner. Astrocytes ensheathe capillaries through polarized end feet that are enriched with aquaporin-4 (Aqp4) proteins, colocalized with inwardly rectified $K^+$ channels (*Nico et al., 2001*; *Nicchia et al., 2004*). Functional coupling of ion and water fluxes plays a critical role in regulating local osmotic equilibrium. During development, radial glia is a neuroepithelial origin with heterogeneous populations that are able to generate neurons, astrocytes, and oligodendrocytes (*Hartfuss et al., 2001*; *Malatesta et al., 2008*; *Qian et al., 2000*; *Price et al., 1995*). In zebrafish, radial glia has long been considered serving an astrocytic role. Radial glia in zebrafish enriches with several key molecular markers for astrocytes and tight junctions while Aqp4+ radial glial processes rarely contact the vasculature (*Grupp et al., 2010*). A more recent article has reported that zebrafish *Glast*-expressing radial glia transform into astrocyte-like cells, displaying dense cellular processes, tiling behavior, and proximity to synapses (*Chen et al., 2020*). Whether end feet of these astrocyte-like cells enwrap capillary blood vessels and resemble mammalian astrocytes warrants future investigations. In agreement with mammalian astrocytes, two independent groups have shown that ablation of pan-glia results in progressive brain hemorrhage (*Johnson et al., 2016*; *Umans et al.,*

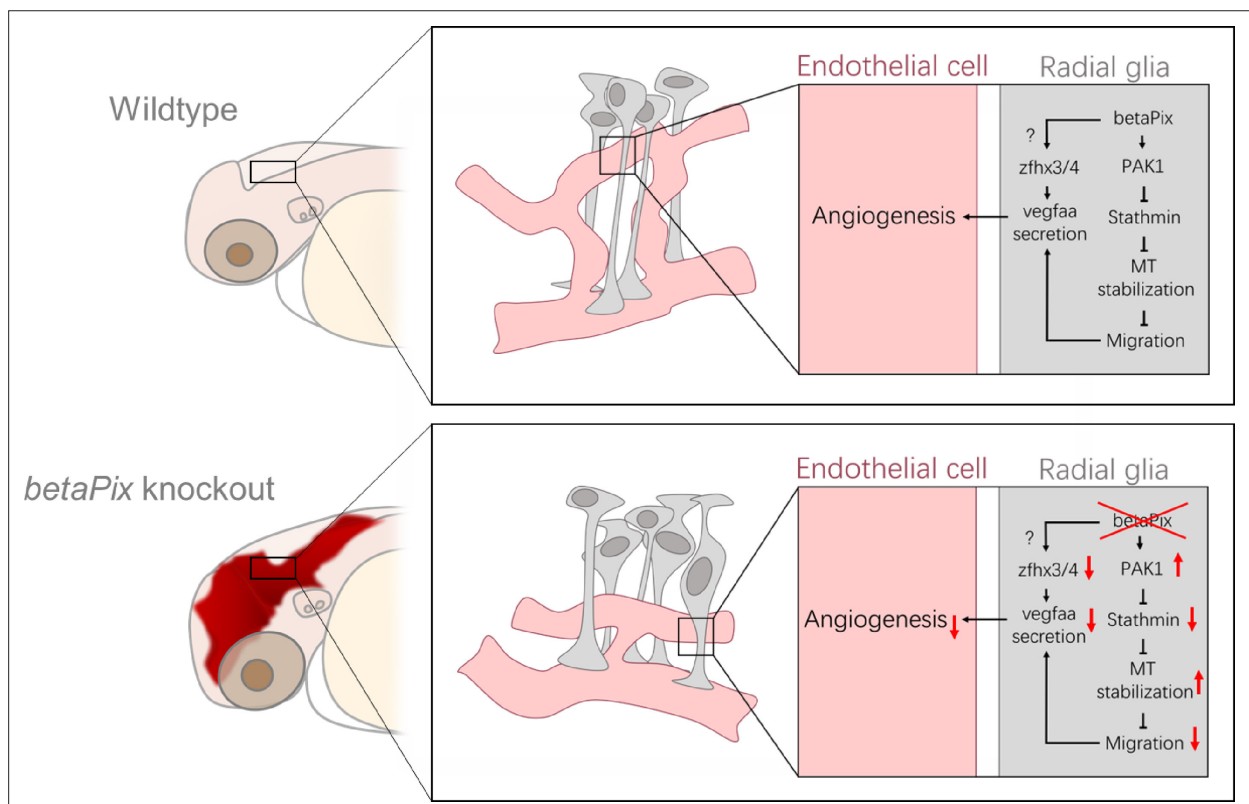

**Figure 7.** Schematic diagram on the function of glial *betaPix* in zebrafish vascular integrity development of the hindbrain. *betaPix* is enriched in glia and regulates the PAK1-Stathmin axis on microtubulin stabilization, thus fine-tuning glial cell migration and their interactions with vascular endothelial cells; and in parallel, betaPix may also regulate Zfhx3/4-Vegfaa signaling in glia, which then modulates angiogenesis during cerebral vessel development and maturation. Deletion of *betaPix* affects glial cell migration and interaction with cerebral endothelial cells. MT, microtubule.

2021) or deficiency of spinal cord arteries development in zebrafish (*Matsuoka et al., 2017*), demonstrating important role of zebrafish glia in BBB formation. Multiple glia/astrocyte-derived signals have been shown to contribute to endothelial barrier properties development. Astrocyte-expressed sonic hedgehog (Shh) binds to hedgehog (Hh) receptors on endothelial cells and contributes to the BBB functions by promoting junctional protein expression and the quiescence of the immune system (*Alvarez et al., 2011*; *Brilha et al., 2017*; *Shi et al., 2024*; *Xing et al., 2020*). Before BBB formation, notochord-derived Shh activity promotes arterial cell fate on developing endothelial cells (*Lawson et al., 2002*). Shh stimulates the production of angiogenic cytokines, angiopoietins, and interleukins, via downstream transcription factor *Gli* or non-canonical pathways crosstalk with iNOS/Netrin-1/PKC, RhoA/Rock, ERK/MAPK, PI3K/Akt, Wnt/β-catenin, and Notch signaling pathways (*Mannan et al., 2023*). By establishing conditional alleles of *betaPix*, we determined the critical roles of betaPix in glia, but not in other brain cells, on its cerebral vessel development and integrity. However, it remains unclear how glial betaPix is related to the above endothelial signaling to regulate endothelial cell function and blood vessel integrity.

It has been known that Pix and Pak participate in multiple signal transduction pathways (*Zhou et al., 2016*). We are interested in the distinctive zebrafish phenotype of the *betaPix* mutants that exhibit severe cranial hemorrhage and hydrocephalus. Previous studies have reported *betaPix-Pak2a* signaling and downstream regulation in focal adhesion essential for vascular stabilization (*Liu et al., 2007*; *Liu et al., 2012*). The experiments reported here establish glial *betaPix*-mediated vascular integrity development via two downstream effectors (*Figures 6M and 7*). IPA-3 has been shown to decrease the level of phosphorylated PAK1 in a rat model of subarachnoid hemorrhage (*Yan et al., 2013*) and presented anti-angiogenic activity in zebrafish (*Jagadeeshan et al., 2017*). Moreover, constitutively active human *Pak1* mRNA administration induced cranial hemorrhage in zebrafish (*Zou et al., 2011*). While previous research focused on *Pak2*, our data suggest that *Pak1* also works with *betaPix* in vascular integrity development.

*Stathmin* was first discovered in a study of leukemia, of which *Stathmins* are rapidly phosphorylated when induced cells undergo terminal differentiation and stopped proliferation (*Braverman et al., 1986*). Subsequent studies have shown downregulated *Stathmin* expressions after stopping proliferation of leukemia cells by chemical reagents (*Luo et al., 1994*). Similar expression changes of *Stathmins* are also found in other tumor cell lines such as breast cancer and ovarian cancer (*Curmi et al., 2000*; *Price et al., 2000*). *Stathmins* have also been associated with the development of the CNS. Systemic *Stathmin* knockout mice develop pan-neural axonopathy upon aging (*Liedtke et al., 2002*). Knockdown or overexpression of *Stathmin* significantly reduced the dendritic growth of Purkinje cells (*Belmont and Mitchison, 1996*). Moreover, *Stathmin* is downregulated with the differentiation of oligodendrocyte progenitor cells in vitro (*Liu et al., 2003a*). In zebrafish, *Stathmin-1* and *Stathmin-2* deficiencies resulted in defective optic nerves development with ectopic branches and axonal adhesion gaps (*Carretero-Rodriguez et al., 2021*). Knockdown or overexpression of *Stathmin-4* led to premature differentiation of dorsal midbrain neural progenitors (*Lin and Lee, 2016*). Systemic knockout of *Stmn4* showed cell cycle arrest in zebrafish retinal progenitor cells (*Jing et al., 2024*). These studies are consistent with our results, suggesting that transcriptionally downregulated *Stathmins* are highly correlated with the impaired glial development in the phenotypic analysis of this study. Previous studies have revealed the role of *betaPix*-d isoform in mouse hippocampal neuron development *via* regulating microtubule stability and PAK-induced *Stathmin* phosphorylation (*Kwon et al., 2020*). It remains to be identified whether deletion of glial- or neuronal-*betaPix* leads to brain hemorrhage. Our study found that *stathmin* knockouts impair development in a variety of cell types in the zebrafish hindbrain. Importantly, hemorrhage and hydrocephalus of *stathmin* knockouts partially phenocopy *betaPix* mutants. In vivo glial-specific *Stmn1b* overexpression confers partial rescue of the glial and neuronal but not vascular developmental defects in *bbh* mutants. Furthermore, ectopic expression of *betaPix* and *Stathmin* alleviates the impaired cell migration and cell shape maintenance in betaPix siRNA glioblastoma cells. Therefore, this work suggests that the *betaPix-Pak1-Stathmin* signaling regulates vascular integrity *via* glial development.

*Zfhx4* administration confers partial protection against *betaPix* depletion-induced hemorrhage, establishing the novel association between the two. However, additional components of this system remain to be discovered. It has been reported that cAMP-response element binding protein (CREB) binds to the CRE consensus site at the ZFHX3 promoter (*Kim et al., 2010*). A variety of protein kinases

modulate CREB activation via phosphorylation at Ser133 (*Gonzalez and Montminy, 1989*), including protein kinase A (PKA), phosphatidylinositol 3-kinase (PI3K)/Akt, mitogen- and stress-activated kinase 1/2 (MSK1/2), and Ca2+/calmodulin-dependent protein kinases and *Pak1* (*Kim and Kaang, 2022*; *Shaywitz and Greenberg, 1999*; *Chung et al., 2018*). Our results demonstrate that *Pak1* inhibition attenuated the majority of phenotypes induced by *betaPix* deficiency. Thus, we speculate *Zfhx3/4* are closely associated with pathogenic mechanisms downstream of *betaPix* signals by modulating expressions of angiogenetic ligands. Further investigations are required to address this hypothesis.

VEGF is the main pro-angiogenic growth factor produced by various cell types that controls multiple vascular development steps, such as proliferation, migration, permeability, and survival via interacting with VEGF receptors on endothelial cells. Inactivation of VEGFA in astrocytes leads to BBB disruption in multiple sclerosis models (*Argaw et al., 2012*; *Chapouly et al., 2015*). In early development of avascular neural tube, VEGF derived from neuroectoderm serve as a central role in inducing tip cells sprouting from the perineural vascular plexus and formation of vasculature lumens from stem cells (*Matsumoto and Mugishima, 2006*; *Jakobsson et al., 2010*). Ablating neuroglia reduces *vegfab* signals and leads to ectopic intersegmental vessels sprouting and vertebral arteries in zebrafish (*Matsuoka et al., 2017*). VEGF signaling orchestrates elaborately with other signaling pathways such as delta-like 4 (Dll4)/Notch signaling (*Liu et al., 2003b*) and activates downstream components including the Ras/Raf/MEK, PI3K/AKT, and p38/MAPK/HSP27 pathways (*Olsson et al., 2006*). Our data suggest glial *betaPix* as an important factor in vascular integrity development. Together with previous studies, *betaPix-Pak1-stathmin* signaling regulates glial growth and differentiation (*Kwon et al., 2020*), and *betaPix* might regulate *Vegfaa* secretion via downstream transcription factors *Zfhx3/4*. Thus, this work presents novel glial *betaPix* signaling in stabilizing and maintaining vascular integrity, and loss of this function causes hemorrhage and subsequent disruption of the BBB (*Figure 7*). However, the molecular mechanism of *betaPix* to *Zfhx3/4*, the sufficiency of *Vegfaa* and the corresponding ligand–receptor interaction between glia and endothelial cells certainly warrant future investigations.

Basement membrane is a non-cellular component consisting of several ECM proteins; serves as hub for structural supports, anchoring, and intercellular communication. Components of the basement membrane are synthesized in surrounding cells with diverse expression patterns (*Xu et al., 2019*). For example, mice with deletion of collagen IV in either brain microvascular endothelial cells (BMEC) or pericytes lead to fully penetrant ICH, while COL4A1 deletion in astrocytes results in mild hemorrhage phenotypes (*Jeanne et al., 2015*). Differential cell-specific expression pattern of laminin isoforms contributes differently to BBB formation (*Halder et al., 2022*). As astrocytic laminins mediate pericyte differentiation and regulate capillary permeability (*Yao et al., 2014*), intercellular interactions with the ECM occur by cell surface integrin receptors. Integrin β1 on endothelial cells is critical during angiogenesis (*Milner and Campbell, 2002*). On the other hand, integrin αvβ8 has glial-specific enrichment which mediates focal adhesion to modulate vascular integrity (*Liu et al., 2012*; *Nishimura et al., 1998*; *McCarty et al., 2005*). Notably, degradation of ECM and focal adhesions at the perivascular space is critical to the hemorrhagic phenotypes of *bbh* mutants (*Liu et al., 2012*; *Yang et al., 2017*). It would be of great interest to determine whether glial *betaPix* accounts for ECM breakdown in *bbh* mutants. Apart from *Vegfaa*, multiple pro-angiogenic growth factors and signaling ligands contribute to effective glial–vascular communication in the perivascular space. Interestingly, ZFHX3 has been reported activating a set of procollagens, integrin, and platelet-derived growth factor receptor β (*Pdgfrb*) genes in response to retinoic acid stimulus, protecting cerebellar neurons from oxidative stress (*Kim et al., 2010*). In the future, it will be interesting to examine whether *Zfhx* signaling modulates adhesion molecules in the pathogenesis of *betaPix* mutants.

## Materials and methods
### Fish maintenance
All zebrafish were raised and handled in accordance with protocol IMM-XiongJW-3, approved by the Institutional Animal Care and Use Committee (IACUC) of Peking University, which is fully accredited by the Association for Assessment and Accreditation of Laboratory Animal Care International (AAALAC). The *bbh*^fn40a mutant was isolated from a large-scale mutagenesis screen of the zebrafish genome at Massachusetts General Hospital, Boston (*Chen et al., 2001*). Tg(*acta2:GFP-Cre*) was purchased from

Xinjia Pharmaceutical Technology Co., Ltd (Nanjing, China); Tg(*neurod:EGFP*) was kindly provided by Dr. Bo Zhang (Peking University, China); Tg(*huC:GFP*) were kindly provided by Dr. Liangyi Chen (Peking University, China); Tg(*kdrl:GFP*) (**Beis et al., 2005**) and Tg(*kdrl:mCherry*) (**Jin et al., 2005**) were described previously.

## Cell culture and transfection

U251 and A172 human glioblastoma cell lines, kindly provided by Dr. Jian Chen at Chinese Institute for Brain Research, Beijing, were obtained from Procell Life Science & Technology Co., Ltd. (Wuhan, China), cultured in high glucose DMEM medium (Hyclone) supplemented with 10% fetal bovine serum and 1% penicillin/streptomycin, at 37°C in 5% $CO_2$ humidified incubator. These cell lines were authenticated (at least once per year) and that they have been regularly tested negative for mycoplasma contamination. The *betaPIX* siRNA (**Connor et al., 2020**) and *STMN1* siRNA (**Jeon et al., 2010**), designed against regions common to all isoforms to ensure knockdown of total *betaPix* and *STMN1* transcripts, were obtained from GenePharma (Shanghai, China). The sequences of the siRNAs are listed in **Supplementary file 2**. The full-length coding cDNA transcripts of human *betaPIX-b* (NM_001354046.2) or *STMN1* (NM_005563.4) were isolated from U251 cDNA library and subcloned into the pcDNA3.1 vector backbone. Transfection was conducted with Lipofectamine 3000 (Invitrogen) according to manufacturer's instructions at a final siRNA concentration of 30 nM.

## mRNA/gRNA synthesis and microinjections

mRNA and gRNA were synthesized as described previously (**Chang et al., 2013**). In brief, linearized pT3TS-nls-zCas9-nls (**Jao et al., 2013**), pXT7-*Cre*, and pXT7-*Zfhx4* plasmid DNA were purified as templates using TIANquick Mini Purification Kit (TIANGEN, DP203, China). Next, in vitro transcription reactions were performed using the mMESSAGE mMACHINE T3 (for pT3TS-nls-zCas9-nls) or T7 (for pXT7-*Cre* and pXT7-*Zfhx4*) kits (Life Technologies) according to manufacturer's instructions. Templates for gRNA were generated by complementary annealing and elongation of two oligos. Forward oligo contained a T7 promoter and gRNA target sequence, and reverse oligo contained the universal gRNA scaffold (**Wu et al., 2018**). The resulting double-stranded DNA served as the templates for in vitro transcription using HiScribe T7 High Yield RNA Synthesis Kit (NEB). The sequences of the gRNAs are listed in **Supplementary file 3**. For global *betaPix* inactivation, *Cre* mRNA with a working concentration of 200 ng/μL was injected into the *betaPix^{ct/ct}* embryos at one-cell stage. For CRISPR-mediated knockouts, 300 ng/μL Cas9 mRNA and 20–50 ng/μL gRNAs were injected into zebrafish embryos at one-cell stage. For *Zfhx4* rescue experiments, *Zfhx4* mRNA with a working concentration of 400 ng/μL was injected into the *bbh^{fn40a}* mutants or in combination with CRISPR-mediated *betaPix* knockout system at one-cell stage. Injected embryos were scored for mutant phenotypes at 36 or 48 hpf.

## Construction of *betaPix* conditional knockout zebrafish

The pZwitch plasmid clone was kindly provided by Dr. Kazu Kikuchi (National Cerebral and Cardiovascular Center Research Institute, Suita, Japan). GSG spacer was added to pZwitch +3 between P2A and TagRFP coding sequences by overlapping primer pairs. Next, a highly efficient gRNA was found from the fifth intron of *betaPix*. Primer pairs were designed to amplify the left and right homologous arms from the gRNA site for 1000 bp and 24 bp with the following modifications: (1) The 5' end of the forward primer of the left homologous arm was added with an *Nhe*1 site and a universal gRNA site which provided an in vivo linearization cleavage site; a *Mlu*1 site to the 5' end of the reverse primer of the left arm. (2) Added *EcoR*1 site to the 5' end of the forward primer of the right homologous arm; added *Xho*1 site and universal gRNA site to the 5' end of the reverse primer of the right homologous arm. Then we cloned the inverted homologous arms into polyclonal sites of the modified pZwitch +3 and purified plasmid DNA with EndoFree Mini Plasmid Kit (Tiangen, DP118). In vivo knock-in system was composed of 300 ng/μL Cas9 mRNA, 50 ng/μL *betaPix*-intron5 gRNA, 50 ng/μL Universal gRNA, and 20 ng/μL donor vector. Microinjected in vivo knock-in system into wild-type embryos at one-cell stage. Fish founders were screened for α-crystallin reporter expression, raised to adulthood, and re-screened for germline transmission and precise knock-in by Sanger sequencing.

## Construction of transgenic reporters

The glial reporter plasmid *gfap*-EGFP (*Bernardos and Raymond, 2006*) was kindly provided by Dr. Jiulin Du (Shanghai Institute for Biological Sciences, Shanghai, China). The *gfap* promoter was used to establish Tg(*gfap:GFP-Cre*) plasmid. The *stmn1b* coding sequences were amplified from wild-type zebrafish cDNA library, which was used to establish Tg(*gfap:GFP-stmn1b*) plasmid.

Transgenic reporters were generated by using *Tol2*-based transgenesis (*Kawakami, 2004*). In brief, 100 ng/µL *Tol2* transposase mRNA in combination with 20 ng/µL donor plasmid DNA was co-injected into the embryos at one-cell stage. Transgenic founders were screened for specific transgenic GFP expression, raised to adulthood, and re-screened for germline transmission.

## Inhibitor treatment

IPA-3 (Proteintech, CM05727) was dissolved in DMSO to form a 10 mM stock, and then diluted with E3 medium to 3 µM in 6-well plates. Zebrafish embryos were treated from 24 to 48 hpf, and then washed with E3 medium for phenotypic analysis.

## *o*-Dianisidine staining

*o*-Dianisidine staining was performed as described previously (*Yang et al., 2017*). In brief, embryos were dechorionated, anesthetized, and incubated in fresh staining solution (0.6 mg/mL *o*-dianisidine, 0.01 M sodium acetate pH 4.5, 0.65% $H_2O_2$, and 40% ethanol) for 20 minutes in the dark. Washed three times with methanol and performed benzyl alcohol/benzyl benzoate tissue clearing before imaging with stereo microscope (Leica, 160F).

## Whole-mount in situ hybridization (WISH)

Whole-mount in situ RNA hybridization was performed as described previously (*Lei et al., 2017*). Anti-sense probes were synthesized using a digoxigenin RNA labeling kit (Roche, 11277073910). Primer sequences for all WISH probes used in this paper are provided in *Supplementary file 4*.

## Quantitative real-time RT-PCR

Total RNA was isolated from embryos using Trizol (Invitrogen) and cDNA was generated using HiScript III RT SuperMix (Vazyme) according to the manufacturer's instruction. Quantitative real-time PCR was performed using Lightcycler (Roche) and ChamQ SYBR qPCR Master Mix (Vazyme). Primer sequences are listed in *Supplementary file 5*. Gene expression level was normalized against GAPDH level.

## Light-sheet fluorescence microscopy imaging

Light-sheet microscopy imaging was performed as described previously (*Pang et al., 2020*). In brief, transgenic zebrafish embryos were collected and maintained at 28.5°C. At the required developmental stages, the embryos were carefully dechorionated, paralyzed with tricaine, and transferred into 1% ultrapure low melting point agarose (16520-050; Invitrogen). The embryo was then drawn into a glass tube in top-down position using a 1 mL syringe with an 18G blunt needle. After agarose coagulation, a wire was inserted from bottom to push the agarose with embryo upwards, removed excess agar until the head area is exposed from the top of the glass tube. Finally, the glass tube was fixed in a sample holder for subsequent imaging.

Imaging was carried out with Luxendo Multi-View Selective-Plane Illumination Microscopy (Bruker). Optical calibrations were adjusted according to the instruction manual.

The optical plates are emitted from two Nikon CFI Plan Fluor 10× W 0.3 NA immersion objectives in opposite directions. The detection is completed by two Olympus 20×1.0 NA immersion lenses. The parameters are set as follows: the green fluorescence channel uses a laser of 488 nm coupled with BP497-554 filter, the red fluorescent channel uses laser 561 nm coupled with BP580-627 filter, with an exposure time of 100 ms, a delay of 11 ms, a line mode of 50 px, and z-Stack interval of 3 µm. The raw data were stored in H5 file format, processed into TIFF format, and merged Multi-View dataset using MATLAB software. After pre-processing, multi-fluorescence channels were merged in ImageJ, and 3D visualization and measurement were performed by Imaris.

## Single-cell RNA sequencing

For knockout samples, 300 ng/µL Cas9 mRNA and a mixture of number 1–4 *betaPix* gRNAs with 50 ng/µL each were co-injected into wildtype embryos at one-cell stage. Injection with equivalent

concentration of Cas9 mRNA with PBS served as siblings. After microinjection, embryos were collected and maintained at 28.5°C. At 24 or 48 hpf, zebrafish were dechorionated, paralyzed, and transferred onto agarose plate. Heads were harvested with dissecting scissors in cold sterile 1X PBS with pooling 200 heads for each group. Heads were then dissociated in 900 µL Accutase cell detachment solution (Sigma-Aldrich, A6964) at 28.5°C for 3 hours and re-suspended by pipetting every 30 minutes. Once digestion was complete, 100 µL FBS was added to cell suspension and centrifuged at 4°C, 500 × g for 3 minutes. The cells were gently re-suspended in cold 500 µL 2.5% FBS in 1X PBS and filtered through a 40 µm strainer. PI and Hoechst33342 were stained for distinguishing living cells.

The single, living cells were sorted by Aria SORP (BD Biosciences) into 1.5 mL tubes. The cell counts and vitality were verified by AOPI staining coupled with an Automated Cell Counter (Countstar BioTech). Around 10,000–12,000 cells were loaded for each group. Single cells were barcoded with Chromium Next GEM Single Cell 3 'Reagent Kits V3.1 kit (10X Genomics, 1000269) in 10X Chromium Controller (10X Genomics). After qualifying by peak shapes, fragment size, and tailing with Fragment Analyzer System kit (Agilent Technologies, DNF-915), single-cell transcriptome libraries were sequenced via Illumina High Throughput Sequencing PE150 (Novogene, Beijing). The sequencing data were analyzed using the CellRanger-6.1.1 (10X Genomics) and mapped to reference genome GRCz11-GRCz11.103. The output numbers of reads in four sample groups are 494,735,089 for ctrl-1d, 615,837,087 for ko-1d, 554,470,209 for ctrl-2d and 583,809,243 for ko-2d.

Low-quality cells were excluded from subsequent analyses under the following conditions: when the number of expressed genes was less than 500, when there were abnormally high counts of UMIs or genes (outliers of a normal distribution), or when the mitochondrial content exceeded 9%. A total of 38,670 cells were qualified for subsequent analyses. Unsupervised clustering was performed using Seurat (version: 4.0.2) with a resolution of 2.5, resulting in 71 cell populations and further annotated into 24 zebrafish major cranial cell types. Differential expression analysis in Seurat v4 was used to identify cluster/cell type markers by Wilcoxon rank-sum test.

## Scratch assay

U251 cells were planted into a culture plate and performed transfection at the desired density. At 24 hours post-transfection, we vertically scratched monolayer cells by using a 200 µL pipette tip. Washed the cells three times with PBS, then replaced them with serum-free culture medium for further culture. Stereo fluorescence microscopy (Leica, 160F) was used to document the scratch size at 0, 18, and 24 hours. Images were processed by ImageJ.

## Immunostaining

For assessing tubulin expression, U251 cells were planted into chamber slides (Saining, 1093000) and transfected at the desired density. At 24 hours post-transfection, washed the cells once on ice with pre-cooled PBS, then fixed with pre-cooled methanol at –20°C for 20 minutes. The fixed cells were permeabilized with pre-cooled acetone at –20°C for 1 minute. Removed acetone, incubated 0.5% BSA/PBS blocking solution at room temperature for 15 minutes. Cells were then incubated with 1:200 anti-α-tubulin mouse monoclonal antibody (EASYBIO, BE0031) at 37°C for 45 minutes. After rinsing once with PBS, cells were incubated with 1:400 Alexa Fluor 488 goat anti-mouse IgG (H+L) secondary antibody (Invitrogen, A11029) at 37°C for another 45 minutes. Washed three times with PBS and sealed with Mounting Medium with DAPI (ZSGB Bio, ZLI-9557). Images were acquired by upright fluorescence microscopy (Leica, DM5000B) at ×40 magnification and processed by ImageJ.

## Statistical analysis

Statistical analysis was performed using GraphPad Prism 6. The statistical significance of differences between the two groups was determined by the independent unpaired Student's t-test. Among three or more groups, one-way ANOVA coupled with Dunnett's test was used. All data are presented as the mean ± SEM. p-Value <0.05 indicates significant, with individual p-values mentioned in the figure/figure legends.

## Acknowledgements

The authors thank Drs. Bo Zhang, Jiulin Du, Jia Li, Liangyi Chen, Kazu Kikuchi, and Jian Chen for providing fish lines, plasmid clones, and human glial cell lines; the members of Dr. Xiong's laboratory

for helpful discussions and technical assistance; and the National Center for Protein Sciences at Peking University, particularly Dr. Liying Du at the Flow Cytometry Core for technical help on the Beckman Coulter MoFlo XDP; and Dr. Hua Liang at National Biomedical Imaging Center, Peking University, for assistance with the Luxendo Multi-View Selective-Plane Illumination Microscopy (Bruker). This work is supported by grants from the National Key R&D Program of China (2019YFA0801602 and 2023YFA1800600); the National Natural Science Foundation of China (32230032 and 31730061).

## Additional information

### Funding

| Funder | Grant reference number | Author |
|---|---|---|
| National Key Research and Development Program of China | 2023YFA1800600 | Jing-Wei Xiong |
| National Natural Science Foundation of China | 31730061 | Jing-Wei Xiong |
| National Key Research and Development Program of China | 2019YFA0801602 | Jing-Wei Xiong |
| National Natural Science Foundation of China | 32230032 | Jing-Wei Xiong |

The funders had no role in study design, data collection and interpretation, or the decision to submit the work for publication.

### Author contributions
Shihching Chiu, Conceptualization, Data curation, Formal analysis, Investigation, Methodology, Writing – original draft; Qinchao Zhou, Xiaojun Zhu, Data curation, Formal analysis, Supervision; Chenglu Xiao, Data curation, Formal analysis; Linlu Bai, Software, Formal analysis; Wanqiu Ding, Software, Formal analysis, Methodology, Writing – review and editing; Jing-Wei Xiong, Conceptualization, Formal analysis, Supervision, Funding acquisition, Investigation, Project administration, Writing – review and editing

### Author ORCIDs
Shihching Chiu ⓘ https://orcid.org/0009-0007-2572-6210
Chenglu Xiao ⓘ https://orcid.org/0000-0002-5880-7830
Jing-Wei Xiong ⓘ https://orcid.org/0000-0001-8438-4782

### Ethics
All zebrafish were raised and handled in accordance with protocol IMM-XiongJW-3, approved by the Institutional Animal Care and Use Committee (IACUC) of Peking University, which is fully accredited by the Association for Assessment and Accreditation of Laboratory Animal Care International (AAALAC).

Reviewer #1 (Public review): https://doi.org/10.7554/eLife.106665.3.sa1
Author response https://doi.org/10.7554/eLife.106665.3.sa2

## Additional files

### Supplementary files
MDAR checklist

Supplementary file 1. merge.seu_res_2.5_logfc0.25_mar.

Supplementary file 2. Primers used for siRNA.

Supplementary file 3. Primers used for guide RNA synthesis.

Supplementary file 4. Primers used for WISH probe synthesis.

Supplementary file 5. Primers used for qRT-PCR.

## Data availability

Single cell RNA-Seq data have been deposited at Dryad.

The following dataset was generated:

| Author(s) | Year | Dataset title | Dataset URL | Database and Identifier |
|---|---|---|---|---|
| Chiu S, Zhou Q, Xiao C, Bai L, Zhu X, Ding W, Xiong J | 2025 | Single-cell RNA-seq of the embryonic zebrafish heads from wild-type siblings and betaPix CRISPR mutants at 1 dpf and 2 dpf | https://doi.org/10.5061/dryad.tb2rbp0g3 | Dryad Digital Repository, 10.5061/dryad.tb2rbp0g3 |

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
