## [Editor Report · eLife Assessment]

This **valuable** article presents findings supported by **solid** data to identify a surprising glia-exclusive function for betapix in vascular integrity and angiogenesis. The article also describes the optimization of a modified CRISPR-based Zwitch approach to generate conditional knockouts in zebrafish.

---

## [Referee Report · Reviewer #1 (Public review)]

The manuscript by Chiu et al describes the modification of the Zwitch strategy to efficiently generate conditional knockouts of zebrafish betapix. They leverage this system to identify a surprising glia-exclusive function of betapix in mediating vascular integrity and angiogenesis. Betapix has been previously associated with vascular integrity and angiogenesis in zebrafish, and betapix function in glia has also been proposed. However, this study identifies glial betapix in vascular stability and angiogenesis for the first time.

The study derives its strength from the modified CRISPR-based Zwitch approach to identify the specific role of glial betapix (and not neuronal, mural or endothelial). Using RNA-in situ hybridisation and analysis of scRNA-Seq data, they also identify delayed maturation of neurons and glia and implicate a reduction in stathmin levels in the glial knockouts in mediating vascular homeostasis and angiogenesis. The study also implicates a betapix-zfhx3/4-vegfa axis in mediating cerebral angiogenesis.

There is both technical (the generation of conditional KOs) and knowledge-related (the exclusive role of glial betapix in vascular stability/angiogenesis) novelty in this work that is going to benefit the community significantly.

However, the study has the following major weaknesses:

(1) The lack of glia-specific rescue of betapix in the global KOs/mutants prevents the study from making a compelling case for the unexpected glial-specific function in vascular development and stability.

(2) Given the known splice-isoform specific function of betapix in haemorrhaging (Liu et al, 2007), at least an expression profile of the isoforms in glia at the relevant timepoints would have further underscored betapix function.

(3) Direct evidence of the status of endothelial cell proliferation/survival deficits, if any, in the glial betapix KOs would have provided a key mechanistic handle. It becomes all the more relevant as Liu et al, 2012 have demonstrated reduced proliferation of endothelial cells in bbh fish and linked it to deficits in angiogenesis.

---

## [Author Response]

The following is the authors’ response to the original reviews.

**Reviewer #1 (Public review):**
The manuscript by Chiu et al describes the modification of the Zwitch strategy to efficiently generate conditional knockouts of zebrafish betapix. They leverage this system to identify a surprising glia-exclusive function of betapix in mediating vascular integrity and angiogenesis. Betapix has been previously associated with vascular integrity and angiogenesis in zebrafish, and betapix function in glia has also been proposed. However, this study identifies glial betapix in vascular stability and angiogenesis for the first time.The study derives its strength from the modified CRISPR-based Zwitch approach to identify the specific role of glial betapix (and not neuronal, mural, or endothelial). Using RNA-in situ hybridization and analysis of scRNA-Seq data, they also identify delayed maturation of neurons and glia and implicate a reduction in stathmin levels in the glial knockouts in mediating vascular homeostasis and angiogenesis. The study also implicates a betapix-zfhx3/4-vegfa axis in mediating cerebral angiogenesis.There is both technical (the generation of conditional KOs) and knowledge-related (the exclusive role of glial betapix in vascular stability/angiogenesis) novelty in this work that is going to benefit the community significantly.While the text is well written, it often elides details of experiments and relies on implicit understanding on the part of the reader. Similarly, the figure legends are laconic and often fail to provide all the relevant details.

Thanks for this reviewer on his/her overall supports on our manuscript. We have now revised the manuscript text and figure legends making them to have all relevant details as much as we can.

Specific comments:(1) While the evidence from cKO's implicating glial betapix in vascular stability/angiogenesis is exciting, glia-specific rescue of betapix in the global KOs/mutants (like those performed for stathmin) would be necessary to make a water-tight case for glial betapix.

We fully agree with the reviewer that it would be ideal to examine glia-specific rescue of betaPix in its global KOs. At the same time, it is difficult to achieve optimal transient expression of betaPix by injecting plasmid clone of gfap:betaPix while it takes long time to establish stable transgenic line gfap:betaPix for rescuing mutant phenotypes. We would like to pursue this line of researches in the future.

(2) Splice variants of betapix have been shown to have differential roles in haemorrhaging (Liu, 2007). What are the major glial isoforms, and are there specific splice variants in the glial that contribute to the phenotypes described?

We agree that it would be important to address whether any specific splice variants in glia contribute to betaPix mutant phenotypes. Previous studies have shown that the isoform a of betaPix is ubiquitously expressed across various tissues, while isoforms b, c, and d are predominantly expressed in the nervous system. In mice, the expression level of isoform betaPix-d is essential for the neurite outgrowth and migration. In the nervous system, we have not assessed glial specific betaPix isoforms directly. Our current data cannot rule out whether specific isoform is involved in its function in glial responses. The Zwitch cassette of betaPix resides on intron 5, thus disrupting all transcripts when Cre is activated. However, we are fully aware of the potential of identifying glial betaPix isoform with direct downstream targets. Further studies to dissect their roles in cerebral vascular development and diseases are part of our future plans.

(3) Liu et al, 2012 demonstrated reduced proliferation of endothelial cells in bbh fish and linked it to deficits in angiogenesis. Are there proliferation/survival defects in endothelial cells in the glial KOs?

We thank the reviewer for highlighting endothelial cell phenotypes in betaPix mutants. We are aware of endothelial cells might directly link to the mutant defects in angiogenesis. We assessed and quantified endothelial migration by measuring the length of developing central arteries, but we did not examine endothelial cell proliferation/survival defects in glial KOs. In our scRNA-seq analysis, the proportion of endothelial cells reduced among betaPix deficiency, indicating that endothelial cell proliferation/survival might decrease in mutants. In this endothelial cell cluster, we found disrupted transcriptional landscape in a set of angiogenic associated genes (Figure 6M). While these analysis highlights altered angiogenic transcriptome profile in endothelial cells of betaPix knockouts, we acknowledge that our study does not directly address proliferation/survival phenotypes in endothelial cells, which warrants future investigations on the role of betaPix in regulating glia-endothelial cell interaction.

**Reviewer #2 (Public review):**
Summary:Using a genetic model of beta-pix conditional trap, the authors are able to regulate the spatio-temporal depletion of beta-pix, a gene with an established role in maintaining vascular integrity (shown elsewhere). This study provides strong in vivo evidence that glial beta-pix is essential to the development of the blood-brain barrier and maintaining vascular integrity. Using genetic and biochemical approaches, the authors show that PAK1 and Stathmins are in the same signaling axis as beta-pix, and act downstream to it, potentially regulating cytoskeletal remodeling and controlling glial migration. How exactly the glial-specific (beta-pix driven-) signaling influences angiogenesis or vascular integrity is not clear.Strengths:(1) Developing a conditional gene-trap genetic model which allows for tracking knockin reporter driven by endogenous promoter, plus allowing for knocking down genes. This genetic model enabled the authors to address the relevant scientific questions they were interested in, i.e., (a) track expression of beta-pix gene, (b) deletion of beta-pix gene in a cell-specific manner.(2) The study reveals the glial-specific role of beta-pix, which was unknown earlier. This opens up avenues for further research. (For instance, how do such (multiple) cell-specific signaling converge onto endothelial cells which build the central artery and maintain the blood-brain barriers?)

We thank this reviewer for his/her overall supports on our work.

Weaknesses:Major:(1) The study clearly establishes a role of beta-pix in glial cells, which regulates the length of the central artery and keeps the hemorrhages under control. Nevertheless, it is not clear how this is accomplished.(a) Is this phenotype (hemorrhage) a result of the direct interaction of glial cells and the adjacent endothelial cells? If direct, is the communication established through junctions or through secreted molecules?

Thanks for this critical question. We attempted to address this issue by performing live imaging using light-sheet confocal microscopy, but failed to achieve sub-cellular resolution. We don’t have data to address this critical issue that warrants future investigations.

(b) The authors do not exclude the possibility that the effects observed on endothelial cells (quantified as length of central artery) could be secondary to the phenotype observed with deletion of glial beta-pix. For instance, can glial beta-pix regulate angiogenic factors secreted by peri-vascular cells, which consequently regulate the length of the central artery or vascular integrity?

Thank the reviewer for this critical point. While we found the major defects of endothelial cell migration quantified by the central artery length, could not rule out the participation of signals from other peri-vascular cells. We fully agree that it will be important to address the cell-type specific relationship by angiogenic factors. Of note, degradation of extracellular matrix and focal adhesion is critical for the hemorrhagic phenotypes of bbh mutants. In a previous published study in our group, we found that suppressing the globally induced MEK/ERK/MMP9 signaling in bbh mutants significantly decreases hemorrhages. Accordingly, we edited a paragraph in the Discussion section on pages 24-25. We plan to continue investigating whether the complex interactions in the perivascular space contribute to vascular integrity disruption, as well as the cross-talks among different cell types during vascular development in these mutants. We believe that our model of glial specific betaPix function will guide us to further study cellular interactions in the follow-up studies.

(c) The pictorial summary of the findings (Figure 7) does not include Zfhx or Vegfa. The data do not provide clarity on how these molecules contribute (directly or indirectly) to endothelial cell integrity. Vegfaa is expressed in the central artery, but the expression of the receptor in these endothelial cells is not shown. Similarly, all other experimental analyses for Zfhx and Vegfa expression were performed in glial cells. More experimental evidence is necessary to show the regulation of angiogenesis (of endothelial cells) by glial beta-pix. Is the Vegfaa receptor present on central arteries, and how does glial depletion of beta-pix affect its expression or response of central artery endothelial cells (both pertaining to angiogenesis and vascular integrity).

Thank this reviewer for pointing out this critical issue. We have now revised the pictorial summary including Zfhx or Vegfa information in Figure 7. The key receptors of VEGF-A ligand are VEGFR-1 and VEGFR-2. In zebrafish, expression of Vegfr-2, as known as kdrl, is well-documented at endothelial cells including the hindbrain central arteries. We fully agree that it would indeed be of great value to assess changes of kdrl expression pattern after betaPix deficiency in vivo. It warrants future investigations to address how the VEGFA-VEGFR2 signaling in endothelial cells is altered in betaPix mutants.

(2) Microtubule stabilization via glial beta-pix, claimed in Figure 5M, is unclear. Magnified images for h-betapix OE and h-stmn-1 glial cells are absent. Is this migration regulated by beta-pix through its GEF activity for Cdc42/Rac?

We have now revised Figure 5M to include magnified images for h-betaPIX and h-STMN1 overexpression groups. It has been shown that there is a positive feedback loop of microtubule regulation consisting of Rac1-Pak1-Stathmin at the cell edge (Zeitz and Kierfeld, 2014 Biophys J.). Previous studies have shown betaPix activates Rac1 through its GEF activity and also regulates the activity of Pak1 via direct binding. As reported by Kwon et al., betaPix-d isoform promotes neurite outgrowth via the PAK-dependent inactivation of Stathmin1. In this work, we did not assess binding activity of betaPix to Rac1 or Pak1. Nevertheless, our data on the rescue experiments via IPA-3 suggest that betaPix deficiency impaired migration through Pak1 signaling.

(3) Hemorrhages are caused by compromised vascular integrity, which was not measured (either qualitatively or quantitatively) throughout the manuscript. The authors do measure the length of the central artery in several gene deletion models (2I, 3C. 5F/J, 6G/K), which is indicative of artery growth/ angiogenesis. How (if at all) defects in angiogenesis are an indication of hemorrhage should be explained or established. Do these angiogenic growth defects translate into junctional defects at later developmental time points? Formation and maintenance of endothelial cell junctions within the hemorrhaging arteries should be assessed in fish with deleted beta-pix from astrocytes.

We appreciate the reviewer’s point and agree that this is a key aspect we need to clarify. To address junctional defects in our model, we re-examined the scRNA-seq data and found mild downregulation of junction protein claudin-5a (cldn5a) levels in the transcriptome analysis of the endothelial cluster (Author response image 1). We agree in principle that single cell RNA sequencing findings should be validated by immunostaining. While we did not measure junctional defects directly in this work, we have previously reported comparable tight junction protein zonula occludens-1 (ZO1) expression between siblings and bbh mutants (Yang et al., 2017 Dis Model Mech). In zebrafish, functionally characterized blood brain barrier (BBB) is only identified after 3 dpf. The lack of mature BBB might be due to the immature status of barrier signature at this developmental stage. Hemorrhage phenotype occurred around 40 hpf, and hematomas would be almost completely absorbed at later stage since most mutants recover and survive to adulthood. Thus future studies are needed to address the junctional characteristics on the cellular and molecular level in later developmental stages of betaPix mutants.

**Author response image 1. sa2fig1:** Violin plots showing cdh5, cldn5a, cldn5b and oclna expression levels in endothelial sub-cluster. ctrl, control siblings; ko, betaPix knockouts (CRISPR mutants); 1d or 2d, 1 or 2 days post fertilization.

(4) More information is required about the quality control steps for 10X sequencing (Figure 4, number of cells, reads, etc.). What steps were taken to validate the data quality? The EC groups, 1 and 2-days post-KO are not visible in 4C. One appreciates that the progenitor group is affected the most 2 days post-KO. But since the effects are expected to be on the endothelial cell group as well (which is shown in in vivo data), an extensive analysis should be done on the EC group (like markers for junctional integrity, angiogenesis, mesenchymal interaction, etc.). Are Stathmins limited to glial cells? Are there indicators for angiogenic responses in endothelial cells?

Thank the reviewer for these critical suggestions. The detailed statements about the quality control steps for 10X sequencing are now provided in the Materials and Methods section. We validate the data quality through multiple steps, including verification of the number of viable cells used in experiment, assessment of peak shapes and fragment sizes of scRNA-seq libraries, confirmation of sufficient cell counts and sequencing reads for data analyses, and implementation of stringent filtering steps to exclude low-quality cells. Stathmins expressions as shown in Violin plots in Figure 4E and stmn1a, stmn1b and stmn4l expressions in UMAP plots in Figure S6C. These expressions are not limited to glial cells but distributed more widely among zebrafish tissues. We would like to point out that despite the small amount, the endothelial cell clusters are presented in Figure 4C with color brown. The proportions of EC groups split by four sample are visualized in Figure S6B and shown significant reduction among betaPix knockouts at 2 dpf, which had similar trend as glial progenitors. In addition, gene ontology analysis identified a set of down-regulated angiogenic genes expression in endothelial cluster (Figure 6M). We realize our interpretation of endothelial cell phenotypes was not sufficiently clear in this work and have now added sentences to the manuscript text on pages 16-17. As noted above, future studies are needed to address how glial betaPix regulates endothelial cell and BBB function.

**Reviewing Editor Comments:**
comments on your manuscript. Addressing comments 1-3 from Reviewer 1 and comment 1 and its subparts from Reviewer 2 (major weaknesses) will significantly improve the manuscript by reinforcing the cell autonomous requirement of betaPix and also gain mechanistic insights. In addition, extensive proofreading and editing of the text, as well as changes to the figure, figure legends, and the discussion as indicated by both reviewers, will improve the readability and clarity of this manuscript.

Thanks for Reviewing Editor on his/her supports on this manuscript. As noted above, we are trying to address the reviewers’ comments using the data we obtained in this work, as well as our plans for future investigations. We have now made extensive proofreading and editing of manuscript text and figure legends for improving the readability and clarity of this manuscript.

**Reviewer #1 (Recommendations for the authors):**
(1) The Discussion is written like an introduction with very little engagement with the data generated in the manuscript. The role of betapix-Pak-stathmin and betapix-zfhx3/4-vegfaa is barely discussed and contextualised vis-à-vis the current knowledge in the field.

We appreciate the reviewer’s critical comments regarding the Discussion section. We have now revised the manuscript text on pages 20-23 to address the role of betapix-Pak-stathmin and betapix-zfhx3/4-vegfaa axis with contributions from this work.

(2) Line 145: "light sheet microscopy" - explain that this was only for experiments involving fluorescence. Currently, it reads as if the data presented in Figures 1D and E are also obtained via light sheet microscopy. E.g., the paragraph starting on line 139 does not say what line was imaged (and what it labels) to reach the conclusions reached. This detail is not there even in the associated figure legend. Similarly, line 153 discusses radial glia, but there is no indication that these were labelled using Tg (GFAP:GFP) except in the figure annotation. There are various instances of such omissions throughout the text, and they should be remedied to indicate what each line is and what it labels, at least in the first instance.

Thank the reviewer for their thoughtful points. In this revised version, we have incorporated more statements of the objectives and methodologies in the text in pages 8-9. We hope that the revised manuscript can better present the data with clarifying methodologies and materials used in this work.

(3) Figure 1E legend: What is the haemorrhage percentage? Is it the number of embryos per experiment showing hemorrhage? Indicate in the text. In the right panel, what is the number of embryos used? Please ensure all numbers (number of embryos, experiments, etc) used to plot any data in the set of figures in the entire manuscript are clearly indicated.

Thank the reviewer for the suggestion. In this revised version, we have incorporated more detailed statements in figures and figure legends in the manuscript to show the numbers of embryos used.

(4) The Discussion section suddenly introduces the blood-brain barrier and extensively discusses it. However, while cerebral haemorrhage can disrupt the BBB and exacerbate the effects of the haemorrhage, this manuscript does not suggest that a weakened BBB is the cause of haemorrhages in betapix mutants. More likely, betapix stabilises and maintains vascular integrity, and loss of this function causes haemorrhaging and subsequent disruption of the BBB. The glial function noted in this study is likely to be distinct from the glial function in BBB development and maintenance. The authors do not show any direct evidence for the latter. These should be shortened, and only relevant aspects facilitating contextualisation of data generated in this manuscript should be retained.

We have now revised the Discussion section to reduce the introduction of blood-brain barrier and add statements according to the suggestions from both reviewers. We hope that the revisions provide a more relevant and balanced discussion.

(5) Is the scratch assay in Figure 5 controlled for differences in cell proliferation among the different manipulations?

We plated the same numbers of cells and cultured them in the same condition. Before conducted scratch assay we replaced medium with serum-free culture medium to reduce the effect from cell proliferation among the different manipulation groups.

(6) In the glioblastoma experiments involving betapix KD, does stathmin RNA/protein decrease? What about Ser 16 phosphorylation (as shown for neurons in Kwon et al, 2020)?

STMN1 RNA was down-regulated by betaPIX deficiency, which was rescued by betaPIX overexpression in glial cells (Author response image 2). These results are similar to those from in vivo analysis (Figure 5A, 5B and S7A). We agree with the reviewer that it would been ideal to examine Ser 16 phosphorylation of Stathmin in our models. However, we believe that our data have established Stathmins function downstream to betaPix.

**Author response image 2. sa2fig2:** qRT-PCR analysis showing that betaPIX over-expression (betaPix OE) rescued STMN1 expression in betaPIX siRNA knockdown (betaPix KD) in U251 cells. Data are presented in mean ± SEM; one-way ANOVA analysis with Dunnett's test, individual P values mentioned in the figure.

(7) How was the rescue of betapix in glioblastoma cells with siRNA-mediated betapix knockdown performed? Is this by betapix-resistant cDNA? Further, no information about isoforms of betapix (both for siRNA-mediated KD and rescue) or stathmin is provided.

As similar to our Zwitch method that disrupting all betaPix transcripts in vivo, the knockdown of human betaPIX were designed to target conserved region of all transcripts in glioblastoma cell lines. And the rescue human betaPIX were obtained from the U251 cDNA library, ideally all isoforms enriched in the glioblastoma cell line would be isolated. The missing details are now provided in the Materials and Methods section, page 26.

(8) It is unclear what the authors' thoughts are on the decrease in stathmin observed and the functional outcome of this decrease. The Discussion could benefit from this.

Thanks. We have now incorporated a new paragraph in the Discussion section at pages 21-22 addressing that down-regulated expression of Stathmins is associated with functional outcome of this decrease.

(9) Zfhx4 mRNA injection is performed on bbh and betapixKO (is this a global or glial KO?) and found to rescue haemorrhaging. While vegfaa mRNA increases, it is formally possible that the rescue is not due to the increase in vegfaa (or that vegfaa is sufficient). Injection of vegfaa mRNA could address this issue.

Zfhx4 mRNA injection was performed on bbh mutants and global betapix knockouts (crispr mutants). To avoid confusion, we have now included a sentence highlighting global knockout mutants used for this rescue experiment. For the second part, we acknowledge that this study cannot definitively prove the necessity of increased vegfaa levels in the rescue experiment. However, our data established Zhfx3/4 as novel downstream effectors to betaPix in cerebral vessel development. And these effects might partly be linked to angiogenic responses regulated by Zhfx3/4. In this revised version, we carefully proposed that Vegfaa signals act downstream of betaPix-Zfhx3/4 axis and highlighted the weakness of our manuscript on not fully investigating sufficiency of Vegfaa in the Discussion section at page 24. We intend to pursue more extensive analysis in our follow-up studies.

(10) A significant part of the manuscript looks at angiogenesis/vascularisation, however, the title of the paper only reflects vessel integrity (which can be distinct from angiogenesis).

Thanks. We have now changed the title to: Glial betaPix is essential for blood vessel development in the zebrafish brain

(11) Line 366: The BBB abbreviation is used without indicating the full form. Perhaps this can be introduced in the preceding sentence.

We have now edited the following sentence: “The maturation hallmark of central nervous system (CNS) vasculature is acquisition of blood brain barrier (BBB) properties, establishing a stable environment ...” in lines 386-387, Discussion section.

(12) Line 371: "rupture" and not "rapture".

We thank the reviewer for pointing out the spelling error, and have now made this correction.

(13) Line 416: "is enriched" instead of "enriches"?

We have now edited as: “...end feet that is enriched with aquaporin-4 ...” in line 411, page 19.

(14) The sentence in lines 121-123 should be simplified.

We have now revised this sentence as the following: “A previous work has shown that bubblehead (bbh^fn40a^) mutant has a global reduction in betaPix transcripts, and bbh^m292^ mutant has a hypomorphic mutation in betaPix, thus establishing that betaPix is responsible for bubblehead mutant phenotypes [10]”.

(15) No mention in the text of what o-dianisine labels.

We have now edited the following sentence: “By using o-dianisidine staining to label hemoglobins, we found severe brain hemorrhages ...” in lines 131-133.

(16) Line 165: Sentence requires improvement. Perhaps "Vascularisation of the central arteries in the zebrafish hindbrain ...".

We have now edited this sentence as: “Vascularisation of the central arteries in the zebrafish hindbrain starts at 29 hpf.” in this revised version (line 176).

(17) Line 184: Why is "hematopoiesis" mentioned? The genesis of blood cells is not tested anywhere in the manuscript.

Thanks. We have now edited this statement as: “IPA-3 treatment had no effect on heamorrhage induction in betaPix^ct/ct^ control siblings.”

(18) Line 222-223: Improve "increasing trends". Perhaps "increased relative proportions". Clarify "progenitors" means neuronal and glial progenitors.

We have now edited this statement: “we found that most neuronal clusters increased relative proportions ...” in this revised version.

(19) Line 232-233: "arrow indicates" - perhaps "indicated by the arrow"? Also, the arrow indicating gfap needs to be mentioned in the Figure S6A legend. Cannot understand what is meant by "as of its enriched gfap".

We have now edited in the text as: “Figure S6A, indicated by the arrow”, and added “Box area and arrow highlighting gfap expressions.” in Figure S6 legend. To avoid confusion, we have revised "as of its enriched gfap" sentence as the following: “We next focused on the progenitor cluster owing to the enriched gfap expression and the significantly reduced numbers of cells in this cluster by betaPix deficiency.”

(20) Line 239 - 240: While the sentence says "... revealed three major categories:", well, more than 3 are mentioned subsequently.

To avoid possible confusion in the text, we have now removed the sub-category examples and presented the data as: “three major categories: epigenetic remodeling, microtubule organizations and neurotransmitter secretion/transportation (Figure 4D).”

(21) Line 252: Stathmins negatively regulate microtubule stability. Why are they referred to as "microtubule polymerization genes stathmins"?

We are thankful to the reviewer for pointing out this error, and we have now made correction in the text as “microtubule-destabilizing protein Stathmins”.

(22) Line 262-265: The citation used to indicate concurrence with mouse data is disingenuous. That study did not show a reduction in stathmin levels upon betapix loss. Rather, it showed an increase in Ser16 phosphorylation on stathmin, which reduces stathmin's microtubule destabilising function. Please elaborate on the difference between the two studies.

We completely agree with the reviewer’s statement that in the cited article, increased Ser16 phosphorylation on stathmin reduces its microtubule destabilising function. While that study did not show a reduction in Stathmin levels, others have shown that transcriptionally downregulated Stathmins are associated with the impaired neuronal and glial development. We have now revised the Discussion section by adding a new paragraph to address the disrupted homeostasis of Stathmins in these previous studies and their possible association with our data. We hope that these changes we made can clarify this issue.

(23) Line 310: While ZFHX3 levels are reduced in betapix mutants and KD in glioblastomas, were ZFHX3 and 4 up- or downregulated in the scRNA-Seq data?

Thanks for this critical point. Indeed, our results showed that ZFHX3 and 4 down-regulated in the glial progenitor cluster in the scRNA-Seq data (Figure S8A) in betaPix knockouts and the FACS-sorted glia cells (Figure S8B).

(24) Line 317: "... betaPix acts upstream to Zfhx3/4-VEGFA signaling in regulating angiogenesis ...". While this is established later, the data at the time of this sentence does not warrant this claim.

We agree with the reviewer’s statement and restated this sentence in the following way: “Zfhx3/4 might act as downstream effector of betaPix.”

**Reviewer #2 (Recommendations for the authors):**
(1) The images shown in 2E/H, 3B, 6F/J can use a schematic that helps readers to understand what to expect or look for. Splitting up the channels may also help in visualizing the vasculature clearly.

Thank the reviewer for these suggestions. In this revised version, we have included schematic diagrams in the figures and incorporated more detailed statements in the legends.

(2) Many times, arrows are pointing to structures (2E/H, 3B), but are not explained clearly (neither in the text nor in the legends). In 3B, the arrow is pointing to a negative space.(3) Legends are minimalistic and do not provide much information. The reader is left to interpret the data on their own.

We apologize for not explaining the figures in enough details. In this revised version, we have now incorporated more detailed statements in the figure legends and have adjusted arrows in all figures.

(4) The text needs heavy proofreading. For example:(a) Line 208- the title does not seem appropriate since the following text does not discuss Stathmins at all, which comes later.

We agree with the reviewer’s statement and restated the title in the following way: “Single-cell transcriptome profiling reveals that gfap-positive progenitors were affected in betaPix knockouts.”

(b) There is no mention of Figure 7 throughout the text.(c) Figure 7 does not include Zfhx or Vegfaa.

Thank the reviewer for pointing out these errors. We have now revised Figure 7 and incorporated it to corresponding paragraphs in the Discussion section.

(5) The discussion seems incoherent in its current state.

We have now revised the Discussion section according to the suggestions from both reviewers. We hope these revisions adequately address your concerns.

(6) Please include some of the following points, if possible, in the discussion.(a) How is GEF activity of Rac/Cdc42 expected to be affected in beta-pix KO fishes?(b) What are the possible different ways the angiogenic pathways merge onto endothelial cells? Or do the authors imagine this process to be entirely driven by glial cells (directly)?

We would like to thank the reviewer for his/her invaluable suggestions. We have now revised the Discussion section and hope that these changes can provide better and more balanced discussion. Since we have no data directly related to GEF activity of Rac/Cdc42 that might be affected in betaPix mutants, as well as have very limited data showing how glial betaPix regulates cerebral endothelial cells and BBB function, we would like to have the Discussion focused on the CRISPR-induced KI and cKO technologies, glial betaPix function and brain hemorrhage, and the putative role of betaPix-Zfhx3/4-VEGF function in central artery development.

References:

Daub, H., Gevaert, K., Vandekerckhove, J., Sobel, A., and Hall, A. (2001). Rac/Cdc42 and p65PAK regulate the microtubule-destabilizing protein stathmin through phosphorylation at serine 16. J Biol Chem 276, 1677-1680. 10.1074/jbc.C000635200.

Kim S, Park H, Kang J, Choi S, Sadra A, Huh SO. β-PIX-d, a Member of the ARHGEF7 Guanine Nucleotide Exchange Factor Family, Activates Rac1 and Induces Neuritogenesis in Primary Cortical Neurons. Exp Neurobiol. 2024;33(5):215-224. doi:10.5607/en24026

Kwon Y, Jeon YW, Kwon M, Cho Y, Park D, Shin JE. βPix-d promotes tubulin acetylation and neurite outgrowth through a PAK/Stathmin1 signaling pathway [published correction appears in PLoS One. 2020 May 13;15(5):e0233327. doi: 10.1371/journal.pone.0233327.]. PLoS One. 2020;15(4):e0230814. Published 2020 Apr 6. doi:10.1371/journal.pone.0230814

Kwon Y, Lee SJ, Shin YK, Choi JS, Park D, Shin JE. Loss of neuronal βPix isoforms impairs neuronal morphology in the hippocampus and causes behavioral defects. Anim Cells Syst (Seoul). 2025;29(1):57-71. Published 2025 Jan 8. doi:10.1080/19768354.2024.2448999

Wittmann, T., Bokoch, G.M., and Waterman-Storer, C.M. (2004). Regulation of microtubule destabilizing activity of Op18/stathmin downstream of Rac1. J Biol Chem 279, 6196-6203.10.1074/jbc.M307261200.

Zeitz, M., and Kierfeld, J. (2014). Feedback mechanism for microtubule length regulation by stathmin gradients. Biophys J 107, 2860-2871.10.1016/j.bpj.2014.10.056.